TOPICAL REVIEW

# Optical mapping of contracting hearts

Vineesh Kappadan[1] , Anies Sohi[1] , Ulrich Parlitz[2], Stefan Luther[2], Ilija Uzelac[3], Flavio Fenton[3], Nicholas S Peters[1], Jan Christoph[4] and Fu Siong Ng[1]

[1] *National Heart and Lung Institute (NHLI), Imperial College London, London, UK*
[2] *Biomedical Physcis Group, Max Planck Institute for Dynamics and Self-Organization, Göttingen, Germany*
[3] *School of Physics, Georgia Institute of Technology, Atlanta, GA, USA*
[4] *Cardiovascular Research Institute, University of California, San Francisco, CA, USA*

Handling Editors: Laura Bennet & Nordine Helassa

The peer review history is available in the Supporting Information section of this article (https://doi.org/10.1113/JP283683#support-information-section).

The Journal of Physiology

**Abstract** Optical mapping is a widely used tool to record and visualize the electrophysiological properties in a variety of myocardial preparations such as Langendorff-perfused isolated hearts, coronary-perfused wedge preparations, and cell culture monolayers. Motion artifact originating from the mechanical contraction of the myocardium creates a significant challenge to performing optical mapping of contracting hearts. Hence, to minimize the motion artifact, cardiac optical mapping studies are mostly performed on non-contracting hearts, where the mechanical contraction is removed using pharmacological excitation–contraction uncouplers. However, such experimental preparations eliminate the possibility of electromechanical interaction, and effects such as mechano-electric feedback cannot be studied. Recent developments in computer vision algorithms and ratiometric techniques have opened the possibility of performing optical mapping

The Journal of Physiology

studies on isolated contracting hearts. In this review, we discuss the existing techniques and challenges of optical mapping of contracting hearts.

(Received 15 September 2022; accepted after revision 27 February 2023; first published online 2 March 2023)

**Corresponding author** F. S. Ng: National Heart and Lung Institute (NHLI), Imperial College London, 4th Floor, ICTEM Building, 72 Du Cane Road, London W12 0NN, UK.    Email: f.ng@imperial.ac.uk

**Abstract figure legend** Combination of excitation ratiometry and motion tracking technique minimizes motion artifacts from optical action potentials (OAPs) in contracting hearts. The isolated contracting heart stained with voltage-sensitive dye is excited with blue and green excitation wavelengths that are rapidly switched in time with each camera frame. The action potential modulated fluorescence emission is then captured on a single camera, separating the frames into odd and even generated optical data corresponding to blue and green excitation or vice versa. Performing motion tracking and computing the ratio between motion-tracked videos significantly reduced the motion artifacts in OAPs as compared to raw signals. Created with biorender.com and published with permission.

## Introduction

Cardiac optical mapping has been extensively used to study and characterize the electrophysiological properties of the heart and to map cardiac arrhythmias (Efimov et al., 2004; Herron et al., 2012). The optical nature of the recording enables high spatial and temporal resolution recordings of electrophysiological data. Cardiac action potentials (APs) and calcium transients in cardiac tissue can be recorded by loading fast-responding voltage- and calcium-sensitive dyes to cardiac tissue, independently or simultaneously (Choi & Salama, 2000; Herron et al., 2012; Uzelac et al., 2022). The dyes exhibit fluorescence, which is the absorption of high energy photons (lower wavelength) and the emission of low energy photons (higher wavelength). For an example of voltage-sensitive dyes, the dye binds to the cardiac cell membrane and undergoes a spectral shift in its excitation and emission (fluorescence) spectrum in response to changes in transmembrane voltage. When the emitted light is appropriately filtered using a bandpass filter, this leads to changes in detected fluorescence intensity during a cardiac action potential that track changes in transmembrane voltage.

Figure 1 illustrates the basic principle of cardiac optical mapping used to record optical action potentials. The cardiac tissue is loaded with a voltage-sensitive dye and then excited with light of wavelengths that lie within its excitation spectrum. The fluorescence emission of the dye that is modulated by transmembrane voltage is captured on a high-resolution photodetector after passing through an emission filter. The commonly used photodetectors are charge-coupled device (CCD), complementary metal oxide-semiconductor (CMOS), and electron multiplying charge coupled device (EMCCD) cameras that have high quantum efficiencies. The emission filter transmits fluorescent light within a certain spectrum of wavelengths. This spectral region generally has a relatively high value of voltage-dependent fractional change in fluorescence ($\Delta F/F$), such that action potential-dependent intensity changes can be easily differentiated from the baseline fluorescence ($F$). However, the $\Delta F/F$ value of currently available voltage-sensitive dyes is very small (Matiukas et al., 2007), with a maximum of around 15%, and hence visualization of action potentials usually requires post-processing techniques such as normalization or background subtraction.

Although multiple voltage-sensitive dyes with different values of $\Delta F/F$ are already available in the market, it is important to note the adverse effects of these dyes in addition to their benefits. For example, the most commonly used ratiometric potentiometric dye, di-4-ANEPPS, was observed to have effects on heart

**Vineesh Kappadan** is a Research Associate at the National Heart and Lung Institute, Imperial College London. He is currently investigating cardiac arrhythmia dynamics in isolated contracting hearts using optical mapping. He was awarded his PhD by University of Göttingen, Germany, for his work on ratiometric optical mapping and marker-free motion tracking of contracting hearts. He completed his master's degree in Applied Optics from Indian Institute of Technology Delhi, and his main area of interests are optics and its application to biology. **Fu Siong Ng** is a Clinical Senior Lecturer in Cardiac Electrophysiology at Imperial College London, and a Consultant Cardiologist. He is a Clinician Scientist and currently leads a programme of research into arrhythmogenic mechanisms alongside performing invasive ablation procedures and implanting pacemakers and defibrillators in patients with heart rhythm disorders. He is also the Programme Director for the intercalated BSc in Cardiovascular Sciences at Imperial College London and Divisional Research Lead at Chelsea and Westminster Hospital NHS Foundation Trust.

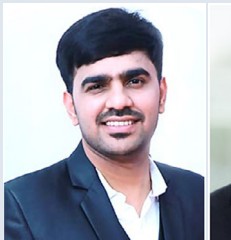
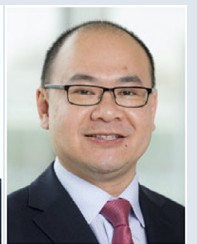

rate (Fialova et al., 2010; Janoušek et al., 2015) in rat, guinea pig and rabbit, conduction velocity (Larsen et al., 2012) in guinea pig, and ischaemia (Ronzhina et al., 2021) in rabbit. However, intracoronary injection of the dye di-4-ANBDQBS (JPW-6033), designed for blood-perfusion optical mapping (Matiukas et al., 2007), showed no signs of cardiac toxicity during continuous monitoring of ECG from baseline to 30 min after dye/solvent injection in pig models. Moreover, heart rate was stable throughout the recording period (Lee et al., 2019).

Measurements based on optical mapping have made important contributions to our understanding and observations of many electrophysiological properties that exhibit spatio-temporal complexity in the heart (Bernus et al., 2007; Cherry & Fenton, 2008; Efimov et al., 1999; Mitrea et al., 2009; Pertsov et al., 1993). Moreover, optical mapping systems are easily adapted to study a wide variety of heart sizes and species (Banville & Grey, 2002; Chattipakorn et al., 2001; Fenton et al., 2009; Handa et al., 2021; Lang et al., 2011; Nanthakumar et al., 2007). Further details about this conventional optical mapping are not within the scope of this paper but can be found in multiple excellent reviews (Attin & Clusin, 2009; Efimov et al., 2004; O'Shea et al., 2020).

## Optical mapping of non-contracting hearts

Until recently, optical mapping studies involving contracting hearts were rarely performed due to the motion artifacts originating from mechanical motion of the cardiac tissue with respect to the detector. The sensitivity of optical mapping studies to motion is very high, because of the small $\Delta F/F$ signal, such that the minimal contractile motion of the heart can cause severe motion artifacts within the optical signal (Christoph et al., 2017; Svrcek et al., 2009). These motion artifacts preclude any reliable analysis of the repolarization properties of the myocardium, and in most cases, also prevent any accurate mapping of the propagation of the action potential wavefront.

Attempts have been made to minimize the motion of the heart without completely suppressing the mechanics, such as applying a moderate level of pressure onto the heart using movable pistons or glass walls (Chen et al., 2000; Laurita et al., 1998; Mandapati et al., 1998). However, these measures alone were often ineffective to eliminate motion artifacts and may cause local ischaemia due to the reduction of coronary flow and myocardial perfusion. The most common method to reduce motion artifacts from optical action potentials (OAPs) is by

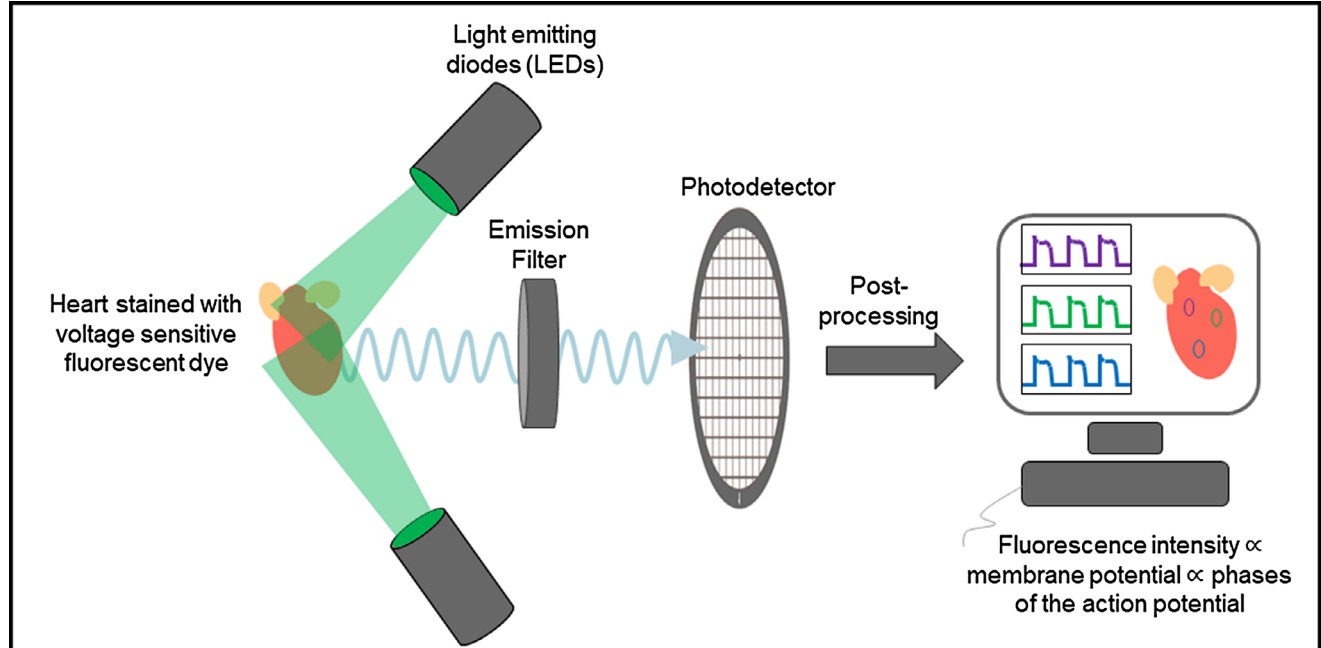

**Figure 1. Principle of optical mapping**
The heart stained with a voltage-sensitive fluorescent dye is illuminated with a specific wavelength of light (e.g. 532 nm for the excitation of Di-4-ANEPPS). The dye emits voltage-dependent fluorescent light, which is first filtered using an emission filter and then captured on a high-resolution photodetector. The recorded intensity on the photodetector is post-processed to obtain optical action potential signals.

administering pharmacological agents to the heart or using low calcium concentrations (Salama & Morad, 1976). The pharmacological excitation–contraction drugs effectively remove any mechanical contraction and uncouple the electrical and mechanical activity of the heart, thus allowing the investigation of the electrophysiological properties of the heart without motion artifact. Three of the most used drugs are 2,3-butanedione monoxime (BDM), also known as diacetyl monoxime, cytochalasin D (Cyto-D) and blebbistatin. BDM and Cyto-D are two non-specific uncouplers such that inhibition of cardiac contractility requires relatively high concentrations. Though the exact mechanism of action of BDM is still unknown, it is a non-competitive inhibitor of myofibrillar ATPase (Fedorov et al., 2007; Sellin & McArdle, 1994). Cyto-D reduces or removes contraction by disrupting actin polymerization (Rueckschloss & Isenberg, 2001), but it is a toxic drug that requires much caution in its use. Blebbistatin, on the other hand, is a specific myosin II inhibitor of actin–myosin interaction and thus abolishes cardiac contraction even at low concentrations (Fedorov et al., 2007; Swift et al., 2021).

Although the excitation–contraction uncouplers are designed to suppress contraction while preserving the electrical activity, BDM and Cyto-D were associated with electrophysiological changes in multiple studies (Banville & Grey, 2002; Kettlewell et al., 2004; Lee et al., 2001; Pitruzzello et al., 2007; Riccio et al., 1999). Blebbistatin, instead, has been considered the electromechanical uncoupler of choice since it was introduced in 2003 (Straight et al., 2003) and is considered to have minimal effects on cardiac electrophysiology (Dou et al., 2007; Fedorov et al., 2007, 2009; Fenton et al., 2008; Hansen et al., 2018; Jou et al., 2010; Lou et al., 2012). Contrastingly, a few studies have recently reported effects of blebbistatin on cardiac electrophysiology (Brack et al., 2013; Kappadan et al., 2020; Lee et al., 2019). The reason behind this disparity could be differences in the metabolic demand and oxygen consumption of contracting hearts as compared to hearts administered with blebbistatin (Kuzmiak-Glancy et al., 2015; Swift et al., 2021). Therefore, the observed differences may not be the direct effects of blebbistatin but may be an indirect consequence of the altered metabolic state of the heart when the contraction is inhibited using an excitation–contraction uncoupler. To date, no studies have shown clear effects of blebbistatin on ion channels responsible for cardiac repolarization. Furthermore, incorrect usage of blebbistatin may lead to blebbistatin precipitation and alterations of cardiac electrophysiology (Swift et al., 2012) and may account for some of the observed electrophysiological changes following blebbistatin administration (Swift et al., 2021).

Even though the excitation–contraction uncouplers allow recording of optical action potentials without motion artifacts, crucial information contained in cardiac contractility and coupled cardiac electro-mechanics is lost. Moreover, electromechanical coupling is bidirectional such that electrical excitation of the cardiac cell membrane causes mechanical contraction via excitation–contraction coupling (ECC) and mechanical stretch on the myocardium can cause electrical excitation through stretch activated channels via a process known as mechano-electric feedback (Franz, 1996, 2000; Taggart, 1996; Zabel et al., 1996), and such effects cannot be studied in a non-contracting heart.

## Optical mapping in contracting hearts

Given the significant limitations of conducting optical mapping experiments in non-contracting hearts paralysed by excitation–contraction uncouplers, there is currently significant interest in developing robust methods to perform cardiac optical mapping in contracting hearts. In this next section, we highlight the three factors that contribute to motion artifacts in the optical action potential and describe novel strategies that facilitate optical mapping of the contracting heart.

## Factors contributing to motion artifacts

Optical mapping studies involving beating hearts are susceptible to motion artifacts, as shown in Fig. 2*Aa*. Optical action potential recorded from contracting hearts are characterized by severe distortion of the baseline and repolarization phase (the fast depolarization phase of the action potential is often largely unaffected). Administration of blebbistatin accordingly eliminates such artifacts (Fig. 2*Ab*). Optical action potentials from contracting hearts are not only affected by the contractile motion of the heart, but the artifact is also affected by the changes in homogeneity of illumination and dye loading. Inhomogeneous illumination and spatio-temporal variation of dye concentration can diminish the quality of recorded OAPs while the heart contracts (Bachtel et al., 2011; Knisley et al., 2000), but importantly, also contribute to motion artifact. The artifact can be caused by differences in fluorescence intensity across different heart regions with inhomogeneous dye concentration and inhomogeneous illumination. As the heart contracts, a camera pixel that was previously recording signals from one region of the heart will now be recording signals from another region, which can have very different fluorescence intensity. These variations in fluorescence can be larger than the intensity change due to action potentials, thus distorting the OAP signal while the heart moves. For example, baseline fluorescence (during diastole) of calcium-sensitive dyes is less as compared to voltage-sensitive dyes, resulting in lesser motion artifacts

in optical calcium transients in comparison with optical action potential. Temporal variation of dye concentration due to photobleaching will also cause motion artifacts by affecting the baseline fluorescence. As an example, the morphologies of OAPs and motion artifacts are completely different at two loci (Fig. 2B, pixel a and pixel b) on the left ventricular surface of a contracting rabbit heart even though the pixel displacements due to the contractile motion are similar. This indicates the possibility of having additional factors contributing to the motion artifacts (Fig. 2C). Furthermore, contraction of cardiomyocytes results in higher membrane density per pixel resulting in a higher signal amplitude and could potentially contribute to motion artifacts in the OAPs.

It is also important to note that motion artifacts observed in optical mapping studies with different voltage-sensitive dyes can also be different as the baseline fluorescence ($F$) and voltage sensitive signal amplitude ($\Delta F/F$) can vary depending on the dye being used. For example, very slow dye washout and signal decay was reported for near infrared dyes di-4-ANBDQPQ and di-4-ANBDQBS in comparison with blue-green excitation dye di-4-ANEPPS (Matiukas et al., 2007). Additionally, light–tissue interactions such as light scattering and absorption are different for different excitation wavelengths and hence may also contribute to motion artifacts.

Hence, the motion artifacts arising from the contracting hearts are due to a combination of all these factors and any robust motion artifact removal technique should address these issues carefully.

### Techniques to reduce motion artifacts

**Motion tracking.** The loss of correspondence between the camera pixel and a specific area of myocardial tissue throughout a cardiac cycle, induced by contraction, is a serious challenge that causes signal distortion (Fig. 2). Adopting motion tracking through the different frames of videos has been shown to minimize motion artifacts in multiple studies. For example, Rohde et al. (2005)

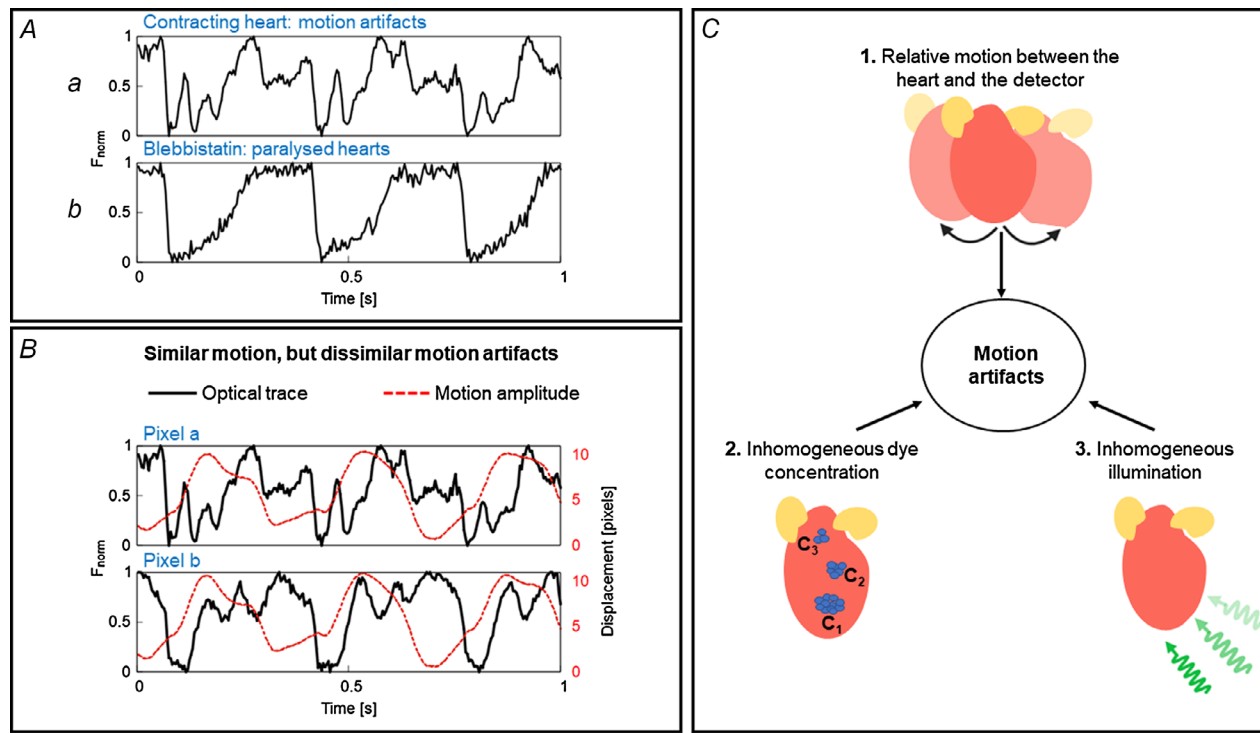

**Figure 2. Causes of motion artifact**
*A*, example traces of optical action potentials (OAPs) from a Langendorff-perfused rabbit heart with and without contractile motion. Optical action potentials in contracting condition are characterized by motion artifacts as indicated by the distortions in the baseline and the repolarization phase of action potentials, whereas OAPs are not distorted when the motion is suppressed using blebbistatin. *B*, variation of motion artifacts and motion amplitude at two different locations (pixel *a* and pixel *b*, $\Delta d_{pixel} = 7$ mm) on the heart. OAPs (continuous black line) from the pixels are distorted differently and hence the motion artifacts are different at these locations despite similar contractile motion amplitudes (dashed red line), indicative of other factors, including inhomogeneous illumination and dye loading, which can contribute to the motion artifacts. *C*, factors contributing to motion artifact: relative motion between the heart and the detector, inhomogeneous illumination, and variations in dye concentration. *A* and *B* are modified from our previous study (Kappadan, 2021).

performed image registration on optical data obtained from rabbit hearts to correct motion artifacts due to global motions but not accounting for local motions. In addition, tracking the motion of the heart using ring-shaped polyethylene markers attached to the epicardial surface has been used to perform simultaneous optical mapping of action potentials and wall motion in isolated perfused pig hearts (Bourgeois et al., 2011). Furthermore, dot-shaped markers and additional geometry cameras were used to reconstruct the epicardial geometry in three dimensions (3D) (Zhang et al., 2016). One major limitation of performing motion tracking using markers is the restriction of spatial resolution of the recording, and interpolation of the data may be required between the markers. To overcome this issue, two-dimensional (2D) marker-free motion tracking that is capable of tracking motion and non-rigid deformation, based on optical flow, has been employed and the efficacy of motion tracking has been demonstrated using synthetic and experimental data (Christoph & Luther, 2018), thus making it possible to perform optical mapping on freely contracting hearts (Christoph & Luther, 2018; Kappadan et al., 2020). Unlike other image registration techniques used for optical mapping data (Khwaounjoo et al., 2015; Rohde et al., 2005), this technique uses locally normalized, contrast enhanced videos for motion tracking. This is a pre-processing step to avoid accidental tracking of electrical activation instead of contractile motion itself. Since motion tracking of optical data can provide information on electrical and mechanical deformations of the heart, it was utilized to study the relationship between the turbulent electrical activity and elasto-mechanical patterns during ventricular fibrillation (Christoph et al., 2018). As an extension, 3D deforming of ventricular wall during sinus rhythm was reconstructed using 2D motion tracking and a multi-camera system, thereby allowing panoramic optical mapping of beating hearts (Christoph et al., 2017).

In this section, below, we describe the basic principle of marker-free 2D motion tracking (Fig. 3). A normal optical mapping recording consists of a series of frames. Each frame is an image with intensity values proportional to the recorded fluorescence intensity. Between the frames, the recorded intensity changes due to motion of the heart and the changes in the action potential, the latter accounting for a small fractional change ($\Delta F/F = 3-10\%$ for ANEP dyes). The motion tracking algorithm computes the geometrical transformation between the reference (R) and test image (T) (Fig. 3A) by registering the test image onto the reference image, such that the difference between test and reference image is minimum. For this, an image frame is selected as the reference image and the geometrical transformation is computed between the reference and test images. For paced data, the reference image is selected as the frame just before the pacing stimulus is delivered, and for Ventricular Fibrillation (VF) data, the reference image is any arbitrary frame. Each test image must be registered with the reference image separately. An important characteristic of this motion-tracking algorithm is that it does not require any markers attached to the heart to assist motion tracking but instead tracks the motion using intensity values between a pair of images (test and reference).

Application of motion tracking to contracting hearts can produce notable reduction in motion artifacts as indicated by the difference image (test minus reference) in Fig. 3*Aa*. The difference image before motion tracking (R − T) is characterized by significant motion artifacts whereas substantial reduction of motion artifacts is visible in the difference image after motion tracking (R − T$_t$). The computed displacement vectors (Fig. 3*Ab*) represent the direction of the contractile motion. Further details about motion tracking algorithms and their validation can be found in the work of Christoph & Luther (2018) as well as in Lebert et al. (2022). Figure 3*B* shows the application of 2D motion tracking where the motion amplitude (pixel displacement) of the heart during fixed frequency pacing and ventricular fibrillation is displayed as motion amplitude maps. As expected, motion amplitude of the contracting heart (no blebbistatin) decreases with increase in blebbistatin concentration. For this heart, contraction was inhibited substantially at a blebbistatin concentration of 2.8 $\mu$M as indicated by a mean motion amplitude value close to zero (0.43 pixels) and in general a blebbistatin concentration <5 $\mu$M completely suppressed the contraction. When the contracting heart is in ventricular fibrillation (VF motion), the mean motion amplitude value is 1.02 pixels (~0.2 mm), indicating that contraction amplitude can be very small during VF, even without blebbistatin, and thus optical mapping has been used without unexplers in those cases (Valderrábano et al., 2001). In some species, such as alligators (Herndon et al., 2021; Jensen et al., 2018) and frogs, blebbistatin does not suppresses contraction, and in those cases, other drugs, low calcium Tyrode solution or motion tracking needs to be used, this last option shown in Fig. 3*C*.

Even though marker-free motion tracking enables performance of optical mapping experiments on isolated contracting hearts, it is important to verify the accuracy of the technique to avoid tracking error and data misinterpretation. The study by Christoph & Luther (2018) validated the robustness and accuracy of 2D motion tracking using experimental and synthetically generated optical mapping videos. They achieved considerable reduction (75–80%) in motion artifacts while comparing the tracked data with simulated ground-truth data. Furthermore, validation of the technique was performed using microelectrode recording as shown in Fig. 3*D*.

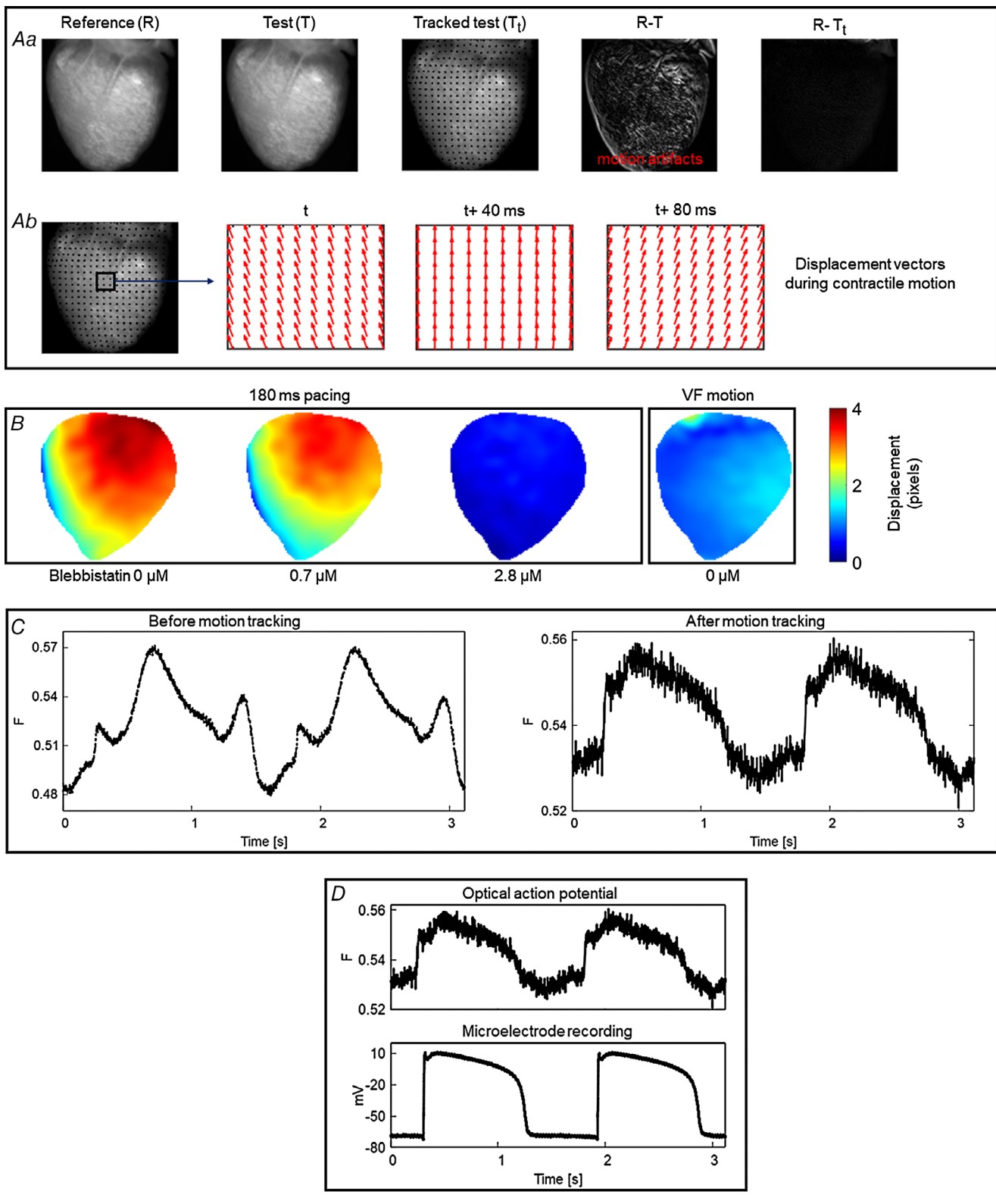

**Figure 3. 2D motion tracking**

*A*, motion tracking algorithm computes the geometrical transformation between test (T) images and a reference (R) image and aligns the test images with the reference image (image registration). *a*, the difference image between the test and the reference image shows significant motion artifacts before motion tracking (R − T). Motion artifacts are significantly reduced in the difference image after motion tracking (R − $T_t$). *b*, displacement

vectors computed via the motion tracking algorithm at three different time points (40 ms intervals) during cardiac contraction. *B*, motion amplitude maps (in pixels) computed using 2D motion tracking during ventricular pacing and fibrillation, showing relatively less motion in VF than during ventricular pacing. Panel modified from our previous study (Kappadan et al., 2020). *C*, example of motion tracking in frog heart, where blebbistatin does not work. Motion tracking significantly reduced the distortion of optical action potentials (motion artifacts). Signals are spatially averaged from 3 × 3 pixels. *D*, validation of motion tracking: comparison of motion tracked optical action potential and a microelectrode action potential recording of a frog heart at 1.5 s period showing good agreement between the signals.

**Excitation and emission ratiometry.** Ratiometric optical mapping techniques (Bachtel et al., 2011; Knisley et al., 2000) are effective for minimizing motion artifact from contracting hearts. Unlike conventional optical mapping, ratiometric techniques record two fluorescence signals that exhibit different changes in fluorescence ($\Delta F$). Taking the ratio between these two signals leads to attenuation of artifacts that are in common. Moreover, this technique diminishes the impact of variations in dye concentrations and illumination artifacts from OAP recordings, which is not addressed by motion tracking algorithms alone (Fig. 2*C*).

There are two principal ratiometry techniques: (i) excitation ratiometry, which uses dual excitation wavelengths to excite the voltage-sensitive fluorescent dye and a single emission band to collect the fluorescent light, which is modulated by transmembrane voltage; and (ii) emission ratiometry, which uses a single excitation wavelength to excite the dye and dual emission bands to collect the fluorescence emission. Excitation ratiometry is relatively easy to implement and can remove common artifacts in OAPs originating from inhomogeneity in dye concentrations across the heart. On the other hand, emission ratiometry requires two detectors that are aligned to image the same field of view of the heart, making it more difficult to implement. However, emission ratiometry can be used to remove inhomogeneous dye loading and illumination artifacts as they are common in OAPs that are recorded on the two detectors.

Excitation ratiometry using pulsed LEDs was first used to record intracellular calcium concentration $[Ca^{2+}]_i$ in cultured rat cardiomyocytes (Fukano et al., 2006, 2008). However, the earliest application of excitation ratiometry to map transmembrane potential in cardiac muscle was demonstrated by Bachtel et al. (2011). They utilized blue (450 ± 10 nm) and cyan (505 ± 15 nm) LEDs to excite Di-4-ANEPPS (1-(3-sulfonatopropyl)-4-[$\beta$-[2-(di-*n*-butylamino)-6-naphthyl] vinyl] pyridinium betaine) and a single bandpass emission filter (585 ± 20 nm) to collect the fluorescence emission from the Langendorff-perfused swine hearts. Similarly, emission ratiometry has been used to reduce motion artifacts in multiple studies (Brandes et al., 1992; Hooks et al., 2001; Hortigon-Vinagre et al., 2016; Knisley et al., 2000; Rohr & Kucera, 1998). For example, Knisley et al. (2000) used two photodetectors

to minimize motion artifacts from isolated rabbit hearts loaded with di-4-ANEPPS and excited by a laser beam of wavelength 488 nm. They observed a reduction in photobleaching-induced drifts and motion artifacts in optical signals while taking the ratio between intensities of two fluorescence emission bands (green and red). Moreover, Hortigon-Vinagre et al. (2016) successfully applied this method to di-4-ANEPPS-loaded human induced pluripotent stem cell-derived cardiomyocytes (hiPSC-CM). Excitation was stimulated with LEDs (470 ± 10 nm) and the fluorescence emission was captured using two photomultipliers at 510−560 nm and 590−650 nm.

Figure 4 shows the principle and application of excitation and emission ratiometry to remove motion artifacts due to the variations in illumination and dye loading. Fig. 4*A* shows the schematic of excitation ratiometry by utilizing the ratiometric property of di-4-ANEPPS. The dye undergoes voltage-dependent shifts in its excitation and emission spectrum. Excitation of the dye with blue and green wavelengths produces action potential-dependent fluorescent changes ($\Delta F$) in opposite directions. Thus, the artifacts that are common in the fluorescence emission of blue and green channels cancel out when taking the ratio between corresponding fluorescence intensities. Example time traces of optical action potentials as depicted in Fig. 4*C* show the reduction in dye loading and illumination artifacts (see limitations of optical mapping in contracting hearts). Figure 4*B* depicts the principle of emission ratiometry, where voltage-dependent fluorescence emission from two emission bands is recorded using two cameras for a single excitation light. The ratio of the recorded intensities from these cameras can reduce motion artifacts due to dye loading and illumination differences. Calculating the ratio of fluorescence intensities for green and red emission bands minimizes motion artifacts (Fig. 4*D*). It is to be noted that the observed reduction in motion artifacts in Fig. 4*C* and Fig. 4*D* are with use of ratiometry technique alone (without motion tracking), and the motion artifacts are not removed completely. This is because ratiometry only corrects for changes in base fluorescence between adjacent moving regions but cannot correct for the interference of action potentials measured between the regions as they move. Therefore, this technique only works if there is little dispersion (i.e. change in AP shape and duration) in the tissue. However, if large

dispersion occurs in small distances, such as during discordant alternans obtained at fast pacing, ratiometry alone will give incorrect AP information. This is even more important for calcium measurements as calcium is not spatially coupled, allowing for large variations at very small distances (Greene et al., 2022; Shiferaw et al., 2005).

**Combination of motion tracking and ratiometry.** Previous optical mapping studies in contracting hearts show that neither motion tracking nor ratiometry alone can remove the motion artifacts completely. To minimize the motion artifacts originating from multiple factors as shown in Fig. 2*C*, motion tracking and emission/excitation ratiometry techniques were

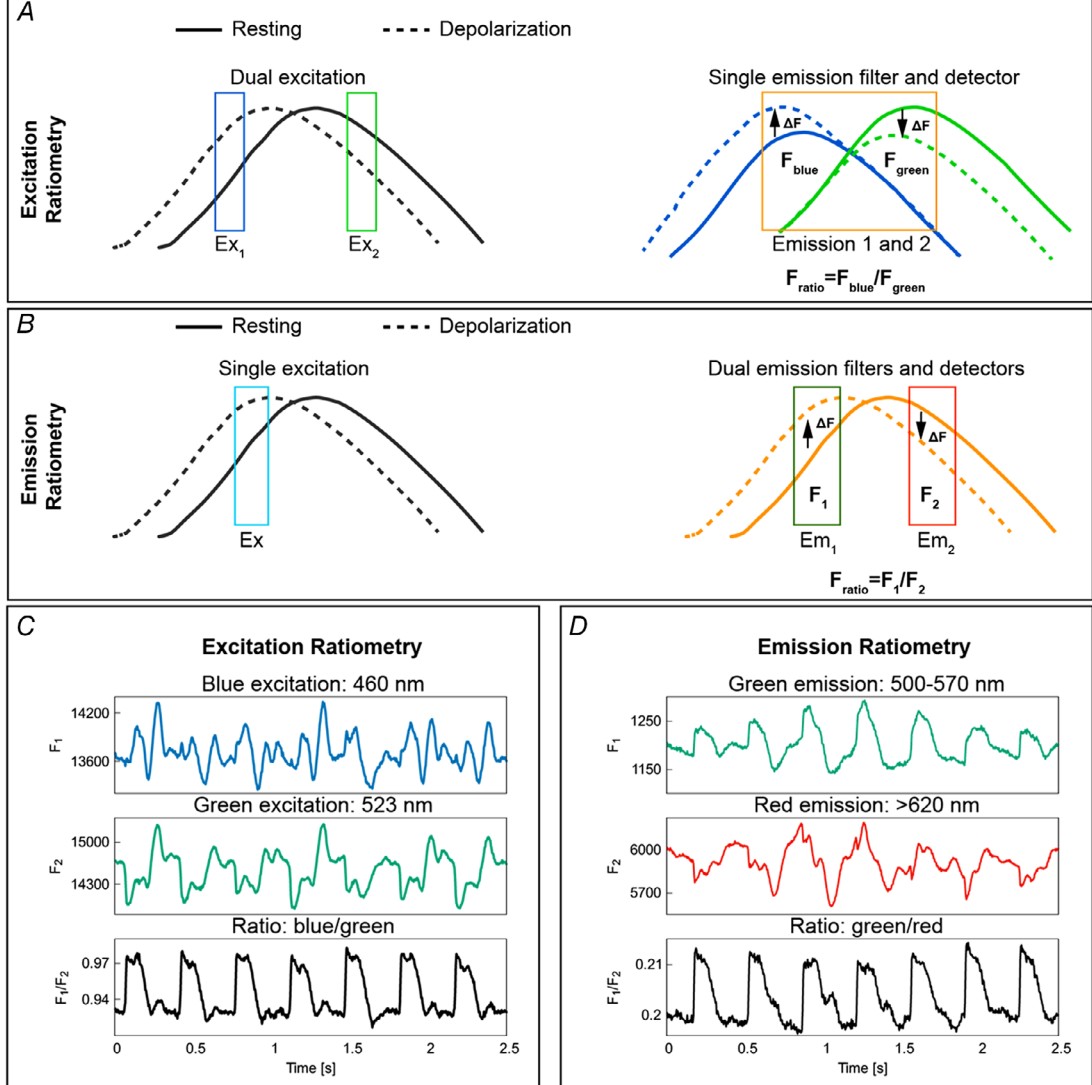

**Figure 4. Excitation and emission ratiometry**

*A*, schematic representation of excitation ratiometry using Di-4-ANEPPS. Excitation ratiometry uses two excitation light sources to excite the dye and a single emission filter–camera combination to collect the action potential-modulated fluorescent light. Exciting the dye with blue and green excitation wavelengths produces positive (dashed blue line above the continuous blue line) and negative (dashed green line below the continuous green line) fractional change in fluorescence ($\Delta F$) during action potential depolarization. The emission filter–camera combination collects emitted fluorescent light for blue and green excitation; taking the ratio of the fluorescence intensities minimizes the motion artifacts originating from illumination and dye loading inhomogeneities. *B*, principle of emission ratiometry. Emission ratiometry uses a single excitation light to excite the dye and two emission wavelength bands (with positive and negative $\Delta F$) to collect the voltage-dependent fluorescence emission. The emission filters transmit the fluorescent light into two cameras that are aligned to look at the same portion of the cardiac tissue. The ratio of the fluorescence intensities of the two cameras reduces dye loading and illumination motion artifacts. *C*, example of OAP signals showing motion artifact reduction using excitation ratiometry. *D*, motion artifact reduction using emission ratiometry.

combined in a few studies. For example, Seo et al. (2010) used fiducial markers to track cardiac tissue deformation and emission ratiometry for action potential mapping. They studied the relationship between strain heterogeneity and initiation of arrhythmia in rabbit ventricular tissue preparations and whole hearts. Bourgeois et al. (2011) used excitation ratiometry for optical mapping and ring-shaped markers for tracking epicardial motion in swine hearts, and epicardial strain was computed using the tracked position of the markers. As an extension to Bourgeois et al.'s method, Zhang et al. (2016) performed motion tracking using dot-shaped fiducial markers attached to the epicardium and hence were able to enhance the spatial resolution by orders of magnitude. Excitation ratiometry was also used here to minimize motion artifacts further. Additionally, they tracked the motion in three dimensions using binocular imaging with additional video cameras and quantified epicardial deformation by computing finite strain fields. More details about electromechanical optical mapping is provided by Nesmith et al. (2020).

In a slightly different approach, we previously used a combination of marker-free motion tracking and excitation ratiometry on isolated rabbit hearts (Kappadan et al., 2020) (Figs 5 and 6). In that study, we measured optical action potential duration ($APD_{50}$ and $APD_{70}$) from Langendorff-perfused contracting hearts and systematically studied APD restitution and ventricular fibrillation dynamics in contracting hearts and compared the properties to that of blebbistatin-uncoupled hearts.

Figure 5 shows the flowchart of our method to perform motion correction technique on Langendorff-perfused contracting hearts. Motion correction involves ratiometry and motion tracking. More specifically, we show the step-by-step processes (Fig. 5*A–E*) involved in the excitation ratiometry, marker-free motion tracking, and the combination of these techniques. In short, we capture optical data for blue and green excitation wavelengths (250 frames each) and perform motion tracking separately from the videos corresponding to each excitation wavelength. The excitation light is switched between camera frames so that odd frames record fluorescent light excited with blue light and the even frames record fluorescent light excited with green excitation light (or vice versa). The ratio between the tracked data (tracked blue and tracked green) is used for further analysis.

The combination of motion tracking and ratiometry (motion correction) significantly reduced motion artifacts and enhanced the quality of OAPs as compared to either of the methods alone (Fig. 6*A*). Furthermore, motion correction techniques can be applied on the entire epicardial surface and properties such as APDs can be measured from a contracting heart with minimal motion artifacts. Figure 6*B* shows an example of motion correction applied on a contracting rabbit heart. Single

pixel time traces from different locations of the contracting heart show severe distortions in optical action potentials due to motion artifacts (dashed lines), whereas motion correction substantially reduced motion artifacts (continuous lines). As depicted in Fig. 6*B*, the morphology of action potential signals heavily varies spatially on the contracting heart, indicating spatial variations in motion artifacts. This implies that the factors contributing to motion artifacts (Fig. 2*C*) are different at these locations. That is, either motion amplitude, or dye concentration or illumination is different at these locations or combinations of these factors are different.

The main benefit of adopting both motion tracking and ratiometry techniques (motion correction) is the elimination of distortions of the repolarization phase of the action potential, which consequently impacts measurements of APDs and cardiac restitution properties. $APD_{80}$ maps, for example, in non-corrected hearts show scattered values of APD across the myocardium, while APD maps on a motion-corrected heart show minimal distortion of APD values across the myocardium (Fig. 7*B*). By using either of the methods alone, residual motion artifacts are clearly visible in the APD maps. Thus, combination of motion tracking and ratiometry is highly recommended for studies involving the repolarization phase of the action potential. On the other hand, the rapid upstrokes of action potentials are minimally affected due to motion artifacts, as shown in the activation time maps (Fig. 7*A*). Activation time for each pixel location is calculated as the time corresponding to the steepest segment of the action potential upstroke ($dF/dt$ max), which can be assigned relatively accurately even in the context of motion artifact (Laughner et al., 2012).

The motion correction technique can also be applied to study the fibrillatory dynamics in contracting hearts. As the amplitude of contractile motion during VF is relatively small, the application of motion correction technique can be highly successful in minimizing motion artifacts. Hence, techniques such as phase analysis can be directly used to visualize VF activation patterns. For example, Christoph et al. (2018) used phase analysis to visualize and characterize VF in motion-tracked pig and rabbit hearts *ex vivo*. Moreover, they computed mechanical phase singularity based on strain rate and showed co-localization of electrical and mechanical phase singularities. We recently compared phase singularity dynamics during VF in isolated contracting rabbit hearts using (i) motion tracking and (ii) combination of motion tracking and ratiometry and showed good agreement between phase maps (Kappadan et al., 2020). This is due to the relatively small motion of the heart during VF. We also investigated the spatial distribution of dominant frequency during VF using Fourier transform-based power spectra analysis, in contracting hearts.

It is also important to discuss the feasibility of ratiometric and marker-free motion tracking techniques to map different animal species and cardiac chambers. The ratiometric optical mapping approach has been successfully used to minimize motion artifacts from OAPs of isolated rabbit (Dumas & Kinisley, 2005; Kappadan et al., 2020; Knisley et al., 2000) and pig (Bachtel et al., 2011; Bourgeois et al., 2011; Lee et al., 2019) whole hearts. Ratiometry was also employed to record action potentials from neonatal rat ventricular myocytes (Chowdhury et al., 2018) and human induced pluripotent stem cell-derived cardiomyocytes (hiPSC-CM) (Hortigon-Vinagre et al., 2016). Moreover, ratiometry utilizing the calcium sensitive dye Indo-1 was used to suppress motion artifacts from intracellular $Ca^{2+}$ concentrations in isolated rat hearts. The marker-free motion tracking has been already used in isolated rabbit (Christoph & Luther, 2018; Christoph et al., 2017, 2018) and pig (Christoph et al., 2018) hearts and a combination of marker-free motion tracking and ratiometry was applied to isolated rabbit hearts (Kappadan et al., 2020). The motion tracking in these studies was performed on the optical mapping

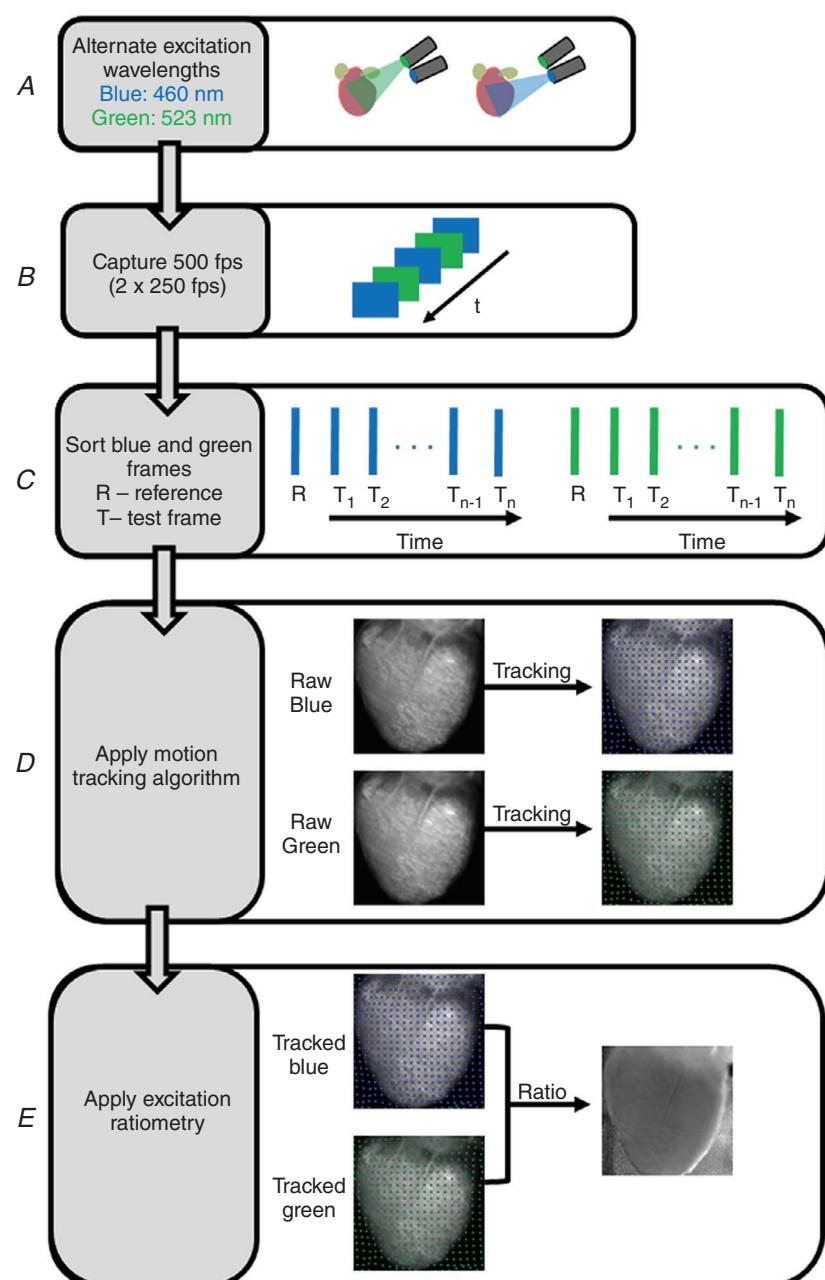

**Figure 5. Combination of excitation ratiometry and motion tracking**
Flowchart representation of motion correction technique in optical mapping studies of beating hearts. *A*, Langendorff-perfused contracting hearts stained with Di-4-ANEPPS are illuminated with blue and green LEDs that are interlaced in time. *B*, the LED switching (500 Hz) is synchronized with the camera (500 fps) such that odd frames of camera record optical action potential for blue excitation and even camera frames of camera record action potentials for green excitation (or vice versa). *C*, sorting of camera frames into odd and even corresponding to blue (raw blue) and green (raw green) excitation. *D*, application of motion tracking algorithm to optical mapping videos of blue and green excitation separately. *E*, the ratio between motion-tracked videos of blue (tracked blue) and green (tracked green) excitation is computed to minimize motion artifacts.

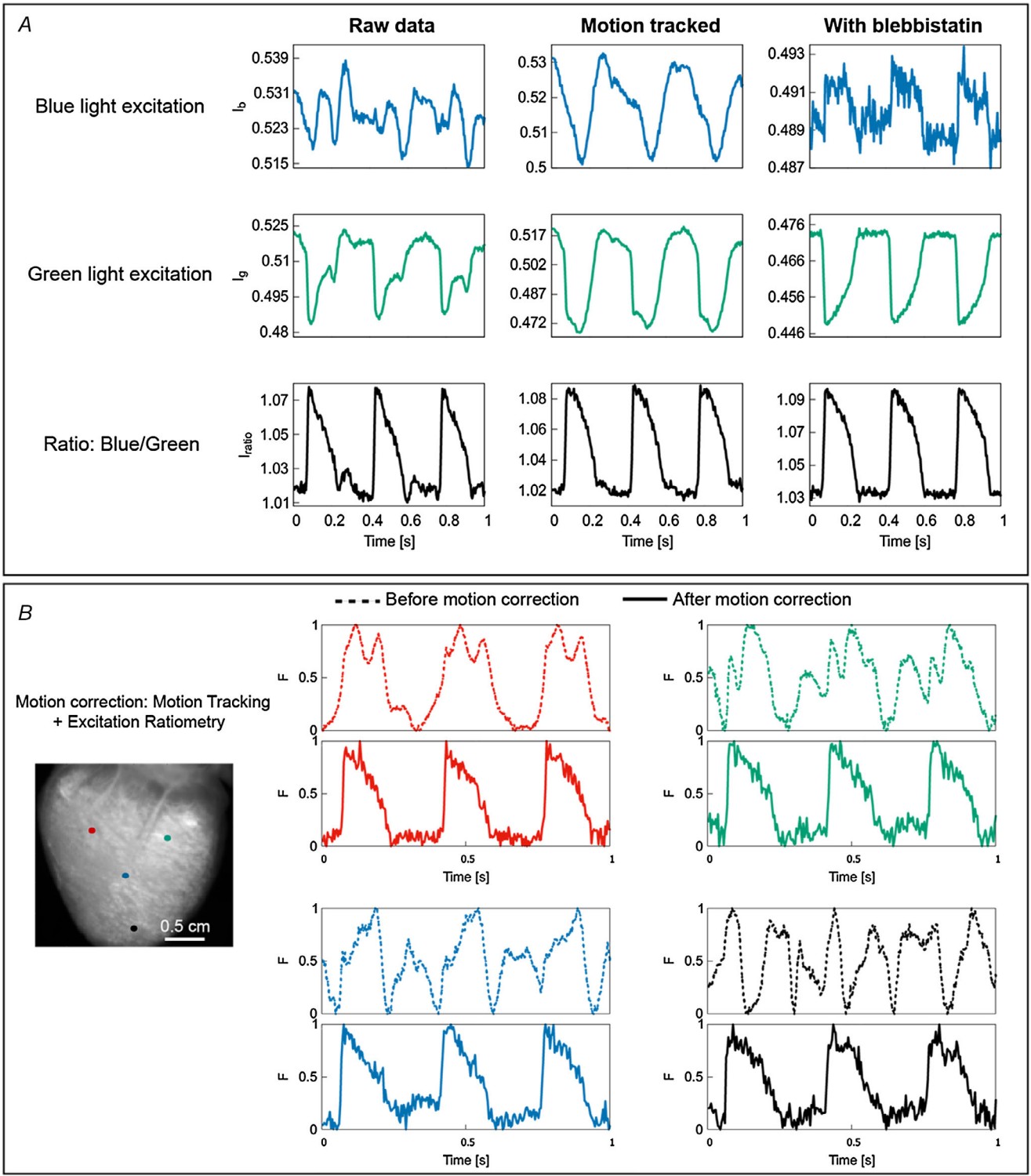

**Figure 6. Motion artifact reduction using combined motion tracking and excitation ratiometry**
*A*, OAP signals from the left ventricle of isolated contracting rabbit heart using motion tracking and excitation ratiometry. Neither motion tracking nor ratiometry removed motion artifacts completely, but a combination of those (bottom row, middle) substantially reduced the artifacts, with signal quality comparable to paralysed-heart experiments using blebbistatin. For this example, spatial average of 3 × 3 pixels was performed. *B*, single pixel OAP traces from four different locations on the left ventricular surface before (dashed) and after (bold) motion correction (motion tracking and ratiometry). A single camera frame is shown on the left for reference. The single pixel OAP is significantly distorted due to motion artifacts when motion correction was not applied.

videos of whole heart (atria and ventricles) with a focus on ventricles during pacing and fibrillation. Recently, GPU-accelerated motion tracking has been successfully applied to hiPSC-CM and isolated hearts of mouse, rabbit and pig (Lebert et al., 2022). However, to our knowledge, no studies have performed motion correction specific to highly trabeculated endocardium or complex atrial structure.

## Limitations of optical mapping in the contracting heart

Next to the significant benefits, optical mapping with contracting hearts still possesses some limitations. Most importantly, the oxygen-carrying capacity of commonly used crystalloid perfusates for *ex vivo* heart preparations is lower than whole blood (Chen et al., 1987; Gaudvel et al., 1985). The metabolic demand and oxygen consumption of the contracting hearts are higher than their electromechanically uncoupled counterparts (Kuzmiak-Glancy et al., 2015). As a result, the contracting hearts are at an increased risk of myocardial hypoxaemia. However, this is not only a limitation of optical mapping but also of the isolated contracting heart experiments in general.

Nevertheless, myocardial oxygenation is a key factor in optical mapping of the contracting hearts, and it should be considered carefully while performing experiments that may cause inadequacy of oxygen supply, such as during high frequency electrical stimulation and recording of the cardiac restitution curve. In such cases, electrophysiological recordings from contracting hearts are also compromised and this can be treated by mechanical arrest using blebbistatin.

Other limitations are mostly related to motion tracking and ratiometry. Despite adopting 2D (i.e. in-plane up-and-down and side-to-side) motion tracking to reduce the impact of motion artifacts, 2D motion tracking cannot account for out-of-plane (i.e. back-and-forth) motions of higher amplitudes, and therefore the capacity of motion tracking algorithms is limited.

As excitation ratiometry involves two different excitation wavelengths that lie within the excitation spectrum of the dye (for example, blue and green), the light intensity variations across the heart tissue with these wavelengths may not be the same. Hence, illumination-related motion artifacts are not common in the recorded OAPs and ratiometry will not remove illumination artifacts completely. Therefore, special care is

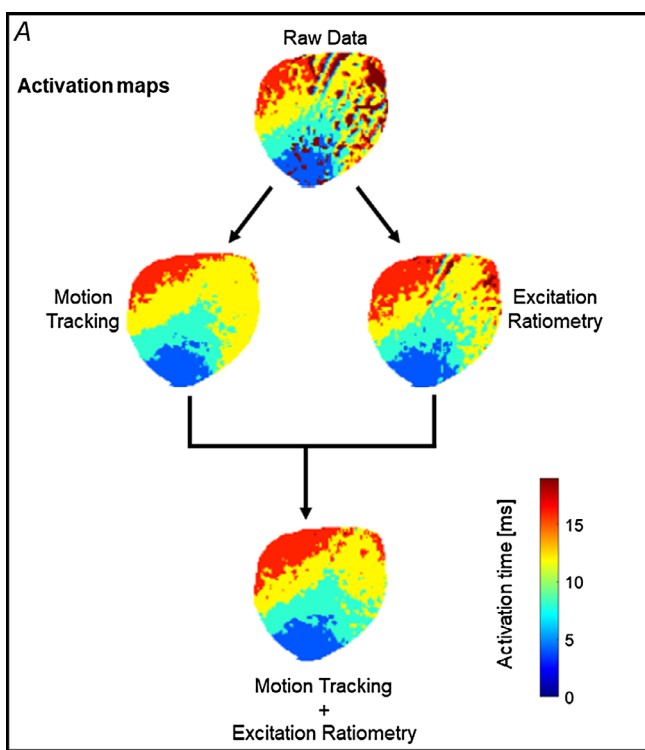
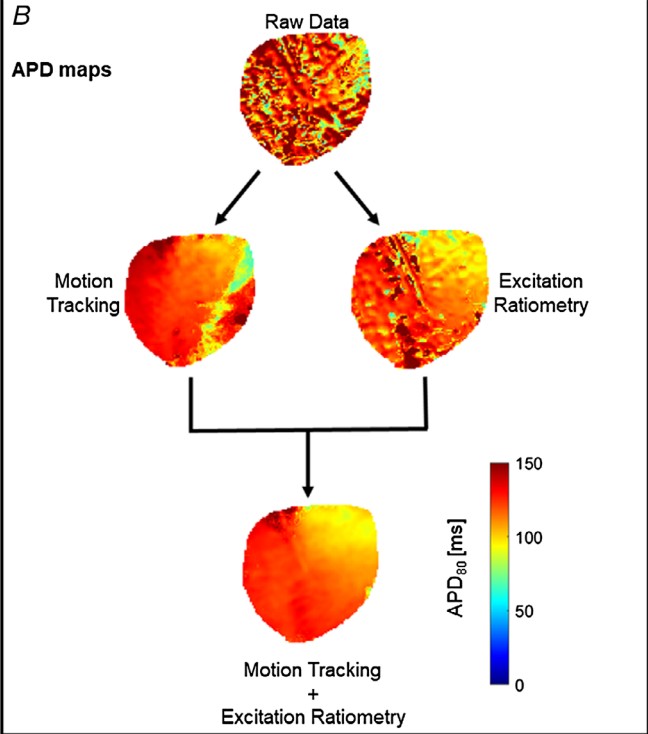

**Figure 7. Comparison of activation and APD maps on a contracting heart with and without motion correction**
*A*, action potential activation maps before and after motion correction. The fast upstroke of action potential is less affected due to motion and thus activation maps are comparable before and after motion correction. *B*, $APD_{80}$ maps before and after motion correction. The combination of motion tracking and ratiometry significantly improved APD measurement as compared to either of the methods alone.

required during the experiment to arrange these excitation light sources to achieve (near) homogeneous intensity across the tissue in the ratio image. The light sources have to be adjusted while looking at the ratio image and the recording can be started once the ratio image shows homogeneous intensity values. Another shortcoming of excitation ratiometry is the loss of temporal resolution of the recording by a factor of 2. Since the recorded camera frames are split into two, corresponding to the blue and the green excitation wavelengths, the temporal resolution of the action potential signals with the excitation ratiometry is half of the camera resolution. For example, 500 fps camera resolution can only produce 250 fps of ratiometric video data.

Due to the aforementioned limitations of motion tracking and ratiometry, the combined motion correction technique is unable to eliminate motion artifacts completely. Accurate measurements of $APD_{90}$ across all pixels on the heart are still difficult to achieve and it is likely that motion-compensated ratiometric OAPs yet contain residual motion artifacts. However, incorporation of motion tracking in 3D can potentially help to reduce the residual artifacts. Moreover, replacing excitation ratiometry with emission ratiometry will in principle solve the issues related to illumination and temporal resolution as it uses only one light source instead of two and hence can minimize motion artifacts further.

### Basic requirements and recommendations for optical mapping of the contracting heart

Here we provide some basic recommendations for performing optical mapping experiments on contracting hearts. From our experience, combining motion tracking and ratiometry can minimize motion artifacts significantly and we recommend using combined techniques for accurate measurement of APD and restitution properties. However, ratiometry or motion tracking alone may be sufficient to compute activation time and activation maps as the action potential upstroke is less impacted by motion. For arrhythmia analysis, such as ventricular fibrillation, where there is reduced motion of the heart, motion tracking may be sufficient (for instance, see Figure 12 from Kappadan et al., 2020), but the combined method is recommended for higher accuracy.

Ratiometric optical mapping of cardiac action potential requires ratiometric voltage-sensitive dyes. The most commonly used dye for ratiometric measurements is di-4-ANEPPS, but studies have been performed using other voltage-sensitive dyes such as di-4-ANBDQBS, di-4-ANEQ(F)PTEA) and di-8-ANEPPS (Chowdhury et al., 2018; Lee et al., 2019). Excitation ratiometry with di-4-ANEPPS requires excitation light with blue and green (or cyan) wavelengths that produces opposite polarity of $\Delta F$, and a single emission filter to collect the fluorescence emission corresponding to both the excitation light. For example, we used blue ($460 \pm 5$ nm) and green ($540 \pm 12.5$ nm) wavelengths for excitation and the fluorescence emission was captured using an EMCCD camera Evolve 128 ($128 \times 128$ pixels, 16-bit dynamic range), Photometrics, Arizona, USA equipped with a bandpass emission filter ($590 \pm 55$ nm) (Kappadan, 2021; Kappadan et al., 2020). It is to be noted that for excitation ratiometry to work well, the excitation light intensities on the heart should be adjusted to achieve a near homogeneous illumination for both wavelengths. For emission ratiometry, we used four blue ($470 \pm 20$ nm) excitation LEDs (Cairn Research, Faversham, UK) and the fluorescence emission is captured on two orthogonally placed CMOS cameras ($128 \times 80$ pixels) using green ($535 \pm 35$ nm, Chroma Technology), Vermont, USA and red ($>620$ nm, Chroma Technology) emission filters. The cameras are carefully aligned to look at the same field of view. Other possible combinations of optics for excitation and emission ratiometry are discussed in the Ratiometry section. 2D motion tracking of the cardiac tissue requires motion tracking (Matlab, The MathWorks, Natick, MA, USA) and visualization (C++) algorithms that can be performed using a single CPU with a computation time of approximately 1 min per video image. However, the computation time can be reduced significantly by using parallel computing and it was recently demonstrated that optical mapping with numerical motion tracking can be performed in real-time (Lebert et al., 2022).

### Summary

This review focuses on the existing techniques for performing optical mapping of isolated contracting hearts. The emergence of computer vision algorithms and numerical motion tracking techniques opens the possibility of performing optical mapping studies in contracting hearts. Motion tracking techniques are highly effective in reducing motion artifacts due to the relative motion between the heart and the camera, whereas ratiometry helps to minimize dye loading and illumination artifacts. Combining these methodologies has shown promise in improving the quality of optically recorded electrophysiological signals and properties, such as action potential duration and cardiac restitution properties. The novel techniques reviewed here can be used to overcome the limitations of conventional optical mapping with paralysed hearts, provide insight into the mechanisms underlying mechano-electrical coupling and cardiac arrhythmia, as well as create experimental conditions involving mechanics to study the effect of myocardial ischaemia and energetics. However, it is important

to note that isolated contracting heart experiments can also be compromised in certain situations owing to the higher oxygen and metabolic demands. In such conditions, excitation–contraction uncoupling using blebbistatin may provide more accurate results and can be preferred. However, on the contrary, there could be effects such as ischaemia underestimation while using blebbistatin due to lower metabolic and oxygen demands. Therefore, there is no single ideal experimental technique to use in all scenarios. Hence, the decision on whether to use contracting heart or EC uncoupled heart depends on the specific research goals. For measurements involving metabolism and cardiac electromechanics including mechano-electric feedback, we recommend using contracting heart optical mapping.

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

## Additional information

### Competing interests

The authors have no conflicts to disclose.

### Author contributions

The experiments were performed in the laboratories of S.L. and F.S.N. F.S.N. and V.K. designed the research. V.K. performed the experiments. V.K. and A.S. prepared the figures and drafted the manuscript. A.S., U.P., S.L., I.U., F.F., N.S.P., J.C. and F.S.N. contributed to experiments and data acquisition. F.S.N., J.C., N.S.P., U.P., S.L., F.F. and I.U. provided critical comments and contributed to the revision of the manuscript. All authors have read and approved the final version of this manuscript and agree to be accountable for all aspects of the work in ensuring that questions related to the accuracy or integrity of any part of the work are appropriately investigated and resolved. All persons designated as authors qualify for authorship, and all those who qualify for authorship are listed.

### Funding

This study was supported by the British Heart Foundation (RG/16/3/32175 and RG/F/22/110078 to F.S.N., N.S.P. and V.K.), Imperial NIHR Biomedical Research Centre funding (to F.S.N.), European Union's Horizon 2020 through the project Advanced BiomEdical OPTICAL Imaging and Data Analysis (BE-OPTICAL) under grant agreement number 675512 (U.P. and S.L.), German Centre for Cardiovascular Research (DZHK e.V.), German Federal Ministry of Education and Research (BMBF, project FKZ 031A147, Go-Bio), the German Research Foundation (DFG, Collaborative Research Centers SFB 1002, Projects B05 and C03 and SFB 937, Project A18) and the Max Planck Society (to S.L.), NIH 1R01HL143450-01 (to F.F.), NSF-FDA-203 789 and NSF 1 762 553 (to F.F.).

### Acknowledgements

The authors would like to thank Danya Agha-Jaffar and Marion Kunze for technical assistance.

### Keywords

action potential duration, motion artifacts, motion tracking, optical mapping, ratiometry, ventricular fibrillation

## Supporting information

Additional supporting information can be found online in the Supporting Information section at the end of the HTML view of the article. Supporting information files available:

**Peer Review History**

