## [Peer Review History · The Journal of Physiology]

Optical mapping of contracting hearts

Vineesh Kappadan, Anies Sohi, Ulrich Parlitz, Stefan Luther, Ilija Uzelac, Flavio Fenton, Nicholas S Peters, Jan Christoph, and Fu Siong Ng

DOI: 10.1113/JP283683

Corresponding author(s): Fu Siong Ng (f.ng@imperial.ac.uk)

The following individual(s) involved in review of this submission have agreed to reveal their identity: Bastiaan J Boukens (Referee #2)

Review Timeline:

Submission Date:	15-Sep-2022
Editorial Decision:	21-Oct-2022
Revision Received:	25-Nov-2022
Editorial Decision:	24-Jan-2023
Revision Received:	09-Feb-2023
Accepted:	27-Feb-2023

Senior Editor: Laura Bennet

Reviewing Editor: Nordine Helassa

Transaction Report:

Dear Dr Ng,

Re: JP-TR-2022-283683 "Optical mapping of contracting hearts" by Vineesh Kappadan, Anies Sohi, Ulrich Parlitz, Stefan Luther, Ilija Uzelac, Flavio Fenton, Nicholas S Peters, Jan Christoph, and Fu Siong Ng

Thank you for submitting your Topical Review to The Journal of Physiology. It has been assessed by a Reviewing Editor and by 2 expert referees and I am pleased to tell you that it is considered to be acceptable for publication following satisfactory revision.

The reports are copied at the end of this email. Please address all of the points and incorporate all requested revisions, or explain in your Response to Referees why a change has not been made.

NEW POLICY: In order to improve the transparency of its peer review process The Journal of Physiology publishes online as supporting information the peer review history of all articles accepted for publication. Readers will have access to decision letters, including all Editors' comments and referee reports, for each version of the manuscript and any author responses to peer review comments. Referees can decide whether or not they wish to be named on the peer review history document.

I hope you will find the comments helpful and have no difficulty in revising your manuscript within 4 weeks.

Your revised manuscript should be submitted online using the links in Author Tasks Link Not Available. This link is to the Corresponding Author's own account, if this will cause any problems when submitting the revised version please contact us.

You should upload:

- A Word file of the complete text (including any Tables);
- An Abstract Figure, (with accompanying Legend in the article file)
- Each figure as a separate, high quality, file;
- A full Response to Referees;
- A copy of the manuscript with the changes highlighted.
- Author profile. A short biography (no more than 100 words for one author or 150 words in total for two authors) and a portrait photograph of the two leading authors on the paper. These should be uploaded, clearly labelled, with the manuscript submission. Any standard image format for the photograph is acceptable, but the resolution should be at least 300 dpi and preferably more.

- A 'Cover Art' file for consideration as the Issue's cover image;
- Appropriate Supporting Information (Video, audio or data set https://jp.msubmit.net/cgi-bin/main.plex?form_type=display_requirements#supp).

To create your 'Response to Referees' copy all the reports, including any comments from the Senior and Reviewing Editors into a Word, or similar, file and respond to each point in colour or CAPITALS. Upload this when you submit your revision.

I look forward to receiving your revised submission.

Yours sincerely,

Professor Laura Bennet
Senior Editor
The Journal of Physiology
<https://jp.msubmit.net>
<http://jp.physoc.org>
The Physiological Society
Hodgkin Huxley House
30 Farringdon Lane
London, EC1R 3AW
UK
<http://www.physoc.org>
<http://journals.physoc.org>

EDITOR COMMENTS

Reviewing Editor:

This Symposium Review focuses on probing interstitial calcium in vivo using optical mapping, a novel technology in constant development. The reviewers found this manuscript well-presented, well-structured, interesting and of great value to the field. However, they highlighted some major concerns and made several suggestions to improve the quality of this work, including adding additional references to avoid any potential bias. Addressing those concerns would greatly enhance the quality and thoroughness of the paper.

REFeree COMMENTS

Referee #1:

This is an interesting review manuscript, which discussed some bases of optical mapping with and without mechanical uncouplers. However, the authors did not provide unbiased overview of some electrophysiological and technical aspects of electro-mechanical coupling and uncoupling. There are several major concerns, which must be resolved to remove bias and improve the value and rigor of this review.

Please also see 'Required Items' below.

COMMENTS

1) Table 1 is biased towards the literature that has similar observations with the authors (ref38). Table 1 didn't acknowledged any studies, which did not show effects of blebbistatin.

Table 1 discussion on pages 6-7 are very misleading. Indeed, BDM and Cyto-D may significantly affects cardiac electrophysiology, but it is not true for widely used uncoupler, blebbistatin.

The issues with inappropriate use of blebbistatin, which can lead to alterations of cardiac electrophysiology were well described by Swift et al. (European Journal of Physiology November 2012, 464 (5):503-512) and more recently reviewed by Swift, Key, Ripplinger and Posnack. AJP 2021; 321:H1005-H1013).

In this "Getting It Right" perspective (Swift et al 2021), the group of optical mapping experts clearly stated that "the incorrect administration of blebbistatin to cardiac tissue can prove detrimental, resulting in erroneous interpretation of optical mapping results."

These studies (ref 36-38) should not blame blebbistatin (BB) unless their kept rigor during ex-vivo experiments and properly administrated BB, since any incorrect use of BB can lead to BB precipitations and un-physiological effects.

In fact, Ref 35 Lou Q, Li W, Efimov IR. AJP 2012 isolated rabbit heart optical mapping study showed (see Figure 1) absence of any effects of BB on ventricular ECG, Monophasic action potential morphology and APD80% during different pacing cycle length.

Furthermore, Ref 26 Fedorov et al HR 2007 study also showed that in rabbit hearts Blebbistatin 5-10 uM did not have any statically significant effects on sinus node, atrial and ventricular transmembrane action potentials parameters including APD 50 and 90% (Table1). BB also did not have effects on cardiac electrical activity, including SR, ECG parameters (PQ, GRS, QT, QTc), atrial, AV nodal and ventricular effective refractory periods, and atrial, AV nodal and ventricular conduction (see Tables 2 and 3).

Fedorov et al, Circ. Res. 2009;104;915-923 ex-vivo canine atria optical mapping study showed that Blebbistatin did not have effects on sinus rhythm and atrial effective refractory period.

More recently, the first in-vivo to ex-vivo canine optical mapping study (Hansen et al Circulation: Arrhythmia and Electrophysiology. 2018;11:e006870) showed that sinus nodal activation, atrial conduction and AF activation pattern are very similar between in-vivo mapping and ex-vivo mapping with blebbistatin.

The authors must acknowledge all these issues with incorrect blebbistatin administration and discuss additional references (see above and below) to provide unbiased overview on blebbistatin as an effective and safe cardiac EM uncoupler in the manuscript and Table 1

2. Table 2 also indicated that blebbistatin decreased the conduction velocity. However, publications (such as the ones above) have shown either no change in conduction velocity or increase in conduction velocity. Ref 38 did not show any data on conduction velocity. Is there a particular reason why such data in authors' work are not discussed while such data from other groups are discussed?

3. The authors have mentioned the limitation that "The metabolic demand and oxygen consumption of the contracting hearts are higher than their electromechanically- uncoupled counterparts⁶⁹. As a result, the contracting hearts are at an increased risk of myocardial hypoxaemia." Considering the contradicting evidence in previous publications that blebbistatin did not impact the conduction velocity and action potential duration, it cannot be ruled out that shorter APD during contraction and even more significant shortening at faster pacing rate is related to the hypoxaemia.

Furthermore, it is crucial to acknowledge that initial ex-vivo baseline conditions (before BB) can be also compromised and cardiac tissues can experience metabolic challenges (e.g. ischemia and activation of IKATP channels) which would lead to shortening of APD and be "treated" by mechanical arrest with BB (see Swift et al 202) As such prolongation of APD (and slowing VF frequency) by BB can be related with restoration of normal physiological APD rather than "direct side" effect of BB on cardiac electrophysiology. So far, no studies showed effects of BB on ion channels responsible for cardiac repolarizations.

4. Lines 141 -144 "We also found that APD₅₀ of contracting rabbit hearts are on average 25 {plus minus} 5% shorter than blebbistatin-uncoupled hearts and VF frequencies were

significantly higher in contracting hearts (15 {plus minus} 4 Hz) as compared to blebbistatin counterparts (9 {plus minus} 2 Hz)."

Induced ventricular arrhythmia in normal rabbit heart (without mechanical uncouplers) is prone to self-termination (see reference below). Can authors describe how long VF usually lasts in their experimental settings?

Manoach, M., et al. "Ventricular self-defibrillation in mammals: age and drug dependence." *Age and ageing* 9.2 (1980): 112-116.

Furthermore, The 15 HZ is a very high VF frequency for the properly coronary artery perfused ex-vivo rabbit heart. Unless authors used additional pharmacological stimulation, 15Hz can be explain by ischemic conditions.

5. The authors have to acknowledge how cardiac motion can effect on optical action potential recordings obtained with different voltage-sensitive dyes (blue-green vs near infrared).

6. For Figures 6-7. It is important to have validation of the method to compare the repolarization (APD) and conduction velocity patterns/values between in-vivo and ex-vivo conditions as well as between different loading approaches during pacing at the same cycle length.

7. The discussed radiometric and motion tracking approaches should be validated by microelectrode and/or monophasic

action potential recordings. Furthermore, the authors should be comparing OAPs, APD and conduction patterns obtained with motion tracking before and after blebbistatin in the same animal.

8. Please discuss and present how VF activation patterns and dominant frequency spatial distribution can be visualized with the motion tracking during in-vivo and ex-vivo conditions.

9. Please discuss side effects of conventional dyes (Di-4 and RH237) vs less toxic near-infrared dyes (ref 37 and Hansen et al, 2018;11:e006870).

10. Furthermore, the authors must clarify species- and cardiac chamber-specific optical mapping approaches including applications of motion tracking/ratiometry and EM uncouplers

Referee #2:

The authors present a thorough review on optical mapping of contracting hearts. This review is well-structured and the necessary literature is well-cited. Please find my minor concerns below.

In the section on electromechanical uncoupler the authors write several times "BDM causes" or "Blebbistatin may affect". Is it certain that these agents are causal? The design of studies investigating this matter are often such that the presence of an uncoupler is compared with the absence of the uncoupler. However, the absence of the uncoupler results in contraction, which causes e.g. high energy demand, pressure changes and stretch. Could it be that many of the differences observed between with and without uncoupler are actually differences caused by beating and non-beating?

In lines 179-182 the authors write: "As the heart contracts, a camera pixel that was previously recording signals from one region of the heart will now be recording signals from another region, which can have very different fluorescence intensity. These variations in fluorescence "can be larger than the intensity change due to action potential, thus distorting the OAP signal while the heart moves." I am convinced that this is the main explanation for movement artifacts in optical action potentials. This because recordings of optical calcium transients show way less movement artifacts than optical action potentials from the same location. The main difference is the baseline fluorescence (during diastole) which is very high for voltage dyes but not for calcium sensors. The authors could consider discussing these observations.

Contraction of myocytes results in higher membrane density per pixel resulting in a higher signal amplitude and could contribute to movement artifact in the optical action potentials.

Line 222. Consider writing "frames" instead of "video frames".

Line 405 "lie" instead of "lies".

In lines 407-409 the authors write "Hence, motion artifacts that originate from the inhomogeneous illumination of the tissue are not common in the recorded OAPs, which makes elimination of illumination artifacts difficult to perform." This sentence is not clear to me. Maybe the authors could adjust to sentence to make it clearer.

The authors describe several motion tracking techniques that can be used for correcting optical signals for movement artifacts. In order to check whether the correction has worked the corrected optical action potential should be compared with a microelectrode recording or, at least, the RT80 should coincide with the upslope of the T-wave of an unipolar electrogram. I encouraged the authors to discuss this matter and use a figure to illustrate this validation.

The authors write almost 1,5 pages about the limitations of optical mapping of contracting hearts. Still in the summary section they advocate the use of this approach. I encourage the authors to be more precise in explaining in which situations optical mapping of contracting hearts is required. But also, to explain when the use of mechano-electrical uncouplers is sufficient or even preferred.

REQUIRED ITEMS:

-Please include an Abstract Figure file, as well as the figure legend text within the main article file. The Abstract Figure is a piece of artwork designed to give readers an immediate understanding of the Review Article and should summarise the main conclusions. If possible, the image should be easily 'readable' from left to right or top to bottom. It should show the physiological relevance of the Review so readers can assess the importance and content of the article. Abstract Figures should not merely recapitulate other figures in the Review. Please try to keep the diagram as simple as possible and without superfluous information that may distract from the main conclusion of the Review. Abstract Figures must be provided by authors no later than the revised manuscript stage and should be uploaded as a separate file during online submission labelled as File Type 'Abstract Figure'. Please ensure that you include the figure legend in the main article file. All Abstract Figures will be sent to a professional illustrator for redrawing and you may be asked to approve the redrawn figure before your paper is accepted.

-The Reference List must be in Journal format https://jp.msubmit.net/cgi-bin/main.plex?form_type=display_requirements#refs

-Please upload separate high quality figure files via the submission form.

-Author profile(s) must be uploaded via the submission form. Authors should submit a short biography (no more than 100 words for one author or 150 words in total for two authors) and a portrait photograph of the two leading authors on the paper. These should be uploaded, clearly labelled, with the manuscript submission. Any standard image format for the photograph is acceptable, but the resolution should be at least 300 dpi and preferably more. A group photograph of all authors is also acceptable, providing the biography for the whole group does not exceed 150 words.

-It is the authors' responsibility to obtain any necessary permissions to reproduce previously published material https://jp.msubmit.net/cgi-bin/main.plex?form_type=display_requirements#use

-Please include a full title page as part of your article (Word) file (containing title, authors, affiliations, corresponding author name and contact details, keywords, and running title).

Response to reviewers: Optical mapping of contracting hearts (JP-TR-2022-283683)

Thank you for the positive outlook on our manuscript entitled “Optical mapping of contracting hearts”, and for providing thoughtful comments and suggestions for improving our manuscript.

We appreciate the opportunity to revise our work, and we are pleased to hereby submit a revised version of our manuscript for possible publication in *The Journal of Physiology*. We have comprehensively addressed all points raised by the reviewers and provide a detailed point-by-point response below. The reviewers’ comments are in ***bold italics*** and are followed by the response to the comments and the revisions undertaken (in blue). We have highlighted the changes made to the manuscript in response to these comments (**Page and Line numbers** below refer to the Redlined document).

As a result of the incisive comments, we believe the manuscript has significantly improved. We hope the revised manuscript is in line with the editor’s expectation as well as those of the reviewers.

EDITOR COMMENTS

Reviewing Editor:

This Symposium Review focuses on probing interstitial calcium in vivo using optical mapping, a novel technology in constant development. The reviewers found this manuscript well-presented, well-structured, interesting and of great value to the field. However, they highlighted some major concerns and made several suggestions to improve the quality of this work, including adding additional references to avoid any potential bias. Addressing those concerns would greatly enhance the quality and thoroughness of the paper.

We thank the Reviewing Editor for the feedback. The required changes have been incorporated into the manuscript to improve the quality of the work.

REFEREE COMMENTS

Referee #1:

This is an interesting review manuscript, which discussed some bases of optical mapping with and without mechanical uncouplers. However, the authors did not provide unbiased overview of some electrophysiological and technical aspects of electro-mechanical coupling and uncoupling. There are several major concerns, which must be resolved to remove bias and improve the value and rigor of this review.

Please also see 'Required Items' below.

We thank the reviewer for their input and have addressed their comments, as listed below. We have now modified the sections on blebbistatin, as suggested, to provide a more balanced view.

COMMENTS

1) Table 1 is biased towards the literature that has similar observations with the authors (ref38). Table 1 didn't acknowledged any studies, which did not show effects of blebbistatin.

Table 1 discussion on pages 6-7 are very misleading. Indeed, BDM and Cyto-D may significantly affects cardiac electrophysiology, but it is not true for widely used uncoupler, blebbistatin.

The issues with inappropriate use of blebbistatin, which can lead to alterations of cardiac electrophysiology were well described by Swift et al. (European Journal of Physiology November 2012, 464 (5):503-512) and more recently reviewed by Swift, Key, Ripplinger and Posnack. AJP 2021; 321:H1005-H1013).

In this "Getting It Right" perspective (Swift et al 2021), the group of optical mapping experts clearly stated that "the incorrect administration of blebbistatin to cardiac tissue can prove detrimental, resulting in erroneous interpretation of optical mapping results."

These studies (ref 36-38) should not blame blebbistatin (BB) unless their kept rigor during ex-vivo experiments and properly administrated BB, since any incorrect use of BB can lead to BB precipitations and un-physiological effects. In fact, Ref 35 Lou Q, Li W, Efimov IR. AJP 2012 isolated rabbit heart optical mapping study showed (see Figure 1) absence of any effects of BB on ventricular ECG, Monophasic action potential morphology and APD80% during different pacing cycle length.

Furthermore, Ref 26 Fedorov et al HR 2007 study also showed that in rabbit hearts Blebbistatin 5-10 uM did not have any statically significant effects on sinus node, atrial and ventricular transmembrane action potentials parameters including APD 50 and 90% (Table1). BB also did not have effects on cardiac electrical activity, including SR, ECG parameters (PQ, GRS, QT, QTc), atrial, AV nodal and ventricular effective refractory periods, and atrial, AV nodal and ventricular conduction (see Tables 2 and 3).

Fedorov et al, Circ. Res. 2009;104;915-923 ex-vivo canine atria optical mapping study showed that Blebbistatin did not have effects on sinus rhythm and atrial effective refractory period.

More recently, the first in-vivo to ex-vivo canine optical mapping study (Hansen et al Circulation: Arrhythmia and Electrophysiology. 2018;11:e006870) showed that sinus nodal activation, atrial conduction and AF activation pattern are very similar between in-vivo mapping and ex-vivo mapping with blebbistatin.

The authors must acknowledge all these issues with incorrect blebbistatin administration and discuss additional references (see above and below) to provide unbiased overview on blebbistatin as an effective and safe cardiac EM uncoupler in the manuscript and Table 1

We thank the referee for the valuable feedback. We have dealt with all of the above comments on blebbistatin here. We have now modified the text to present a more balanced view on blebbistatin. **Table 1** has been modified with the addition of studies that showed no effect of blebbistatin in order to present an unbiased view of the topic. Specifically, we have added the following five studies (Fedorov et al, 2007; Lou et al, 2012; Fenton et al, 2008; Dou et al, 2007; Jou et al, 2010).

Additionally, the discussion relating to **Table 1** has been modified by adding information on studies that did not show effects on electrophysiology (**Pages 6-7, lines 149-158**) as follows:

“For example, Fedorov et al observed no significant effect of blebbistatin on cardiac electrophysiological properties such as pacemaker activity, conduction and repolarization in rabbit hearts (Fedorov et al., 2007). More specifically, 5-10 μ M of blebbistatin did not show any effects on sinus node and atrial and ventricular action potential parameters including APD₅₀ and APD₉₀. Additionally, studies by Lou et al showed no effect of blebbistatin on MAP, MAPD₈₀ and ECG in isolated rabbit hearts (Lou et al., 2012). Fedorov et al also showed no effect of blebbistatin on sinus rhythm and atrial effective refractory period in a study on canine atria (Fedorov et al., 2009). Administration of blebbistatin also did not alter cardiac action potentials in equine hearts (Fenton et al., 2008), mouse cardiomyocytes (Dou et al., 2007), and zebrafish hearts (Jou et al., 2010).”

Moreover, to provide an unbiased overview, the information on metabolic demand of contracting hearts and incorrect blebbistatin administration are also included (**Page 7, lines 171-183**) as follows:

*“However, we also observed morphology changes in OAPs indicating possible ischemia during high frequency electrical stimulations (**Figure 4D**, Kappadan et al., 2020). This is because metabolic demand and oxygen consumption of contracting hearts are higher as compared to hearts administered with blebbistatin and hence contracting hearts are more prone to ischemia especially while pacing at higher frequencies. Therefore, the observed differences may not be the direct effects of blebbistatin but may be an indirect consequence of the altered metabolic state of the heart when the contraction is inhibited using an excitation-contraction uncoupler. It is therefore unclear if blebbistatin significantly affects cardiac electrophysiology. To date, no studies have shown clear effects of blebbistatin on ion channels responsible for cardiac repolarization. Furthermore, incorrect usage of blebbistatin may leads to blebbistatin precipitation and alterations of cardiac electrophysiology (Swift et al., 2012) and may account for some of the observed electrophysiological changes following blebbistatin administration (Swift et al., 2021).”*

2. Table 2 also indicated that blebbistatin decreased the conduction velocity. However, publications (such as the ones above) have shown either no change in conduction velocity or increase in conduction velocity. Ref 38 did not show any data on conduction velocity. Is there a particular reason why such data in

authors' work are not discussed while such data from other groups are discussed?

Thank you for this comment. The publication mentioned here (Lee *et al.*, 2019 (ref 37 in old version)) might be confusing as it shows comparison of conduction velocity under different conditions, such as between pre- and post-blebbistatin, as well as between *ex vivo* and *in vivo*. They observed an increase in conduction velocity in *in vivo* conditions as compared to *ex vivo* (**Figure 6D**). However, a decrease in conduction velocity was observed with blebbistatin administration (**Figure 3D**).

We (Kappadan *et al.*, 2020, ref 38 in old version) did not show any data on conduction velocity as the study was mainly focussed on APD and VF dynamics. However, to provide an unbiased overview, we modified **Table 1** to include studies that did not show any effects of blebbistatin on conduction velocity.

3. The authors have mentioned the limitation that "The metabolic demand and oxygen consumption of the contracting hearts are higher than their electromechanically- uncoupled counterparts⁶⁹. As a result, the contracting hearts are at an increased risk of myocardial hypoxaemia." Considering the contradicting evidence in previous publications that blebbistatin did not impact the conduction velocity and action potential duration, it cannot be ruled out that shorter APD during contraction and even more significant shortening at faster pacing rate is related to the hypoxaemia.

Furthermore, it is crucial to acknowledge that initial ex-vivo baseline conditions (before BB) can be also compromised and cardiac tissues can experience metabolic challenges (e.g. ischemia and activation of IKATP channels) which would lead to shortening of APD and be "treated" by mechanical arrest with BB (see Swift et al 202) As such prolongation of APD (and slowing VF frequency) by BB can be related with restoration of normal physiological APD rather than "direct side" effect of BB on cardiac electrophysiology. So far, no studies showed effects of BB on ion channels responsible for cardiac repolarizations.

Thank you for the feedback. The limitation section is modified to acknowledge the possibility of compromising baseline recordings of contracting hearts due to the metabolic challenges and possibility of treatment using blebbistatin (**Page 16, lines 476-478**) as follows:

"In such cases, electrophysiological recordings from contracting hearts are also compromised and can be treated by mechanical arrest using blebbistatin."

The effects of metabolic challenges are also included while discussing observed effects of blebbistatin versus contracting heart (**Page 7, lines 171-180**) as follows:

*"However, we also observed morphology changes in OAPs indicating possible ischemia during high frequency electrical stimulations (**Figure 4D**, Kappadan *et al.*, 2020). This is because metabolic demand and oxygen consumption of contracting hearts are higher as compared to hearts administered with blebbistatin and hence*

contracting hearts are more prone to ischemia especially while pacing at higher frequencies. Therefore, the observed differences may not be the direct effects of blebbistatin but may be an indirect consequence of the altered metabolic state of the heart when the contraction is inhibited using an excitation-contraction uncoupler. It is therefore unclear if blebbistatin significantly affects cardiac electrophysiology. To date, no studies have shown clear effects of blebbistatin on ion channels responsible for cardiac repolarization."

4. Lines 141 -144 "We also found that APD50 of contracting rabbit hearts are on average 25 {plus minus} 5% shorter than blebbistatin-uncoupled hearts and VF frequencies were significantly higher in contracting hearts (15 {plus minus} 4 Hz) as compared to blebbistatin counterparts (9 {plus minus} 2 Hz)."

Induced ventricular arrhythmia in normal rabbit heart (without mechanical uncouplers) is prone to self-termination (see reference below). Can authors describe how long VF usually lasts in their experimental settings?

Manoach, M., et al. "Ventricular self-defibrillation in mammals: age and drug dependence." Age and ageing 9.2 (1980): 112-116.

Furthermore, The 15 HZ is a very high VF frequency for the properly coronary artery perfused ex-vivo rabbit heart. Unless authors used additional pharmacological stimulation, 15Hz can be explain by ischemic conditions.

Thank you for your comments. The VF episodes in our experimental settings were sustained for 10-20 minutes in contracting and blebbistatin administered hearts. However, we found it difficult to induce VF on blebbistatin administered hearts as compared to contracting conditions.

5. The authors have to acknowledge how cardiac motion can effect on optical action potential recordings obtained with different voltage-sensitive dyes (blue-green vs near infrared).

Thank you for your feedback. A paragraph (**Page 9, lines 233-240**) is included to acknowledge the effect of different voltage sensitive dyes in contributing motion artifacts:

"It is also important to note that motion artifacts observed in optical mapping studies with different voltage-sensitive dyes can also be different as the baseline fluorescence (F) and voltage sensitive signal amplitude ($\Delta F/F$) can vary depending on the dye being used. For example, very slow dye washout and signal decay was reported for near infrared dyes di-4-ANBDQPQ and di-4-ANBDQBS in comparison with blue-green excitation dye di-4-ANEPPS (Matiukas et al., 2007). Additionally, light-tissue interactions such as light scattering and absorption are different for different excitation wavelengths and hence may also contribute to motion artifacts."

6. For Figures 6-7. It is important to have validation of the method to compare the repolarization (APD) and conduction velocity patterns/values between in-

vivo and ex-vivo conditions as well as between different loading approaches during pacing at the same cycle length.

Thank you for your comments. The validation of motion tracking using a microelectrode recording is shown in **Figure 3D**. We are afraid that we are unable to use a microelectrode record to validate motion tracking for data shown in Figure 6-7 as the data was obtained previously in a different laboratory – Also, as this is a review rather than an original article, we feel it is not within the scope of this review. However, *Christoph & Luther, 2018* has validated the technique of 2D marker-free motion tracking using experimental and synthetic optical mapping videos.

7. The discussed radiometric and motion tracking approaches should be validated by microelectrode and/or monophasic action potential recordings. Furthermore, the authors should be comparing OAPs, APD and conduction patterns obtained with motion tracking before and after blebbistatin in the same animal.

Thank you for your suggestions. An extra subfigure (**Figure 3D**) is added to show the validation of motion tracking using a microelectrode recording. The comparison of OAPs, APD and conduction patterns shown in Figure 6-7 are in the same heart.

8. Please discuss and present how VF activation patterns and dominant frequency spatial distribution can be visualized with the motion tracking during in-vivo and ex-vivo conditions.

Thank you for the feedback. A paragraph (**Page 15, lines 441-453**) is now added to discuss about VF activation patterns and dominant frequency spatial distribution using the motion correction technique:

“The motion correction technique can also be applied to study the fibrillatory dynamics in contracting hearts. As the amplitude of contractile motion during VF is relatively small, the application of motion correction technique can be highly successful in minimizing motion artifacts. Hence, techniques such as phase analysis can be directly used to visualize VF activation patterns. For example, Christoph et al used phase analysis to visualize and characterize VF in motion tracked pig and rabbit hearts (Christoph et al., 2018) ex vivo. Moreover, they computed mechanical phase singularity based on strain rate and showed co-localization of electrical and mechanical phase singularities. We recently compared phase singularity dynamics during VF in isolated contracting rabbit hearts using (a) motion tracking and (b) combination of motion tracking and radiometry and showed good agreement between phase maps (Kappadan et al., 2020). This is due to the relatively small motion of the heart during VF. We also investigated the spatial distribution of dominant frequency during VF using Fourier transform based power spectra analysis, in contracting hearts.”

9. Please discuss side effects of conventional dyes (Di-4 and RH237) vs less toxic near-infrared dyes (ref 37 and Hansen et al, 2018;11:e006870).

Thank you for your suggestion. A paragraph (**Pages 4-5, lines 80-87**) is now

dedicated in the introduction to discuss the side effects of conventional versus less toxic near infrared dyes:

“Although multiple voltage-sensitive dyes with different values of $\Delta F/F$ are already available in the market, it is important to note the adverse effects of these dyes in addition to their benefits. For example, the most commonly used ratiometric potentiometric dye, di-4-ANEPPS was observed to have effects in heart rate (Fialova et al., 2010; Janoušek et al., 2015) in rat, guinea pig and rabbit, conduction velocity (Larsen et al., 2012) in guinea pig and ischemia (Ronzhina et al., 2021) in rabbit. However, the effects of near infrared dyes such as di-4-ANBDQPQ (JPW-6003) and di-4-ANBDQBS (JPW-6033), designed for blood perfused optical mapping (Matiukas et al., 2007), have not been reported or studied in detail.”

10. Furthermore, the authors must clarify species- and cardiac chamber-specific optical mapping approaches including applications of motion tracking/ratiometry and EM uncouplers

Thank you for your feedback. Species and cardiac chamber dependency of optical mapping of contracting hearts are discussed in a separate paragraph (**Pages 15-16, lines 454-463**) as:

“It should be noted that, as the marker-free motion tracking technique reviewed here is based on the recorded intensity on the cameras, it should work for all animal species and cardiac chambers for similar or higher spatial resolutions. Ratiometric techniques should also work in all species and chambers if optical components are properly selected (see requirements and recommendation section). For example, marker-free motion tracking has been already used in isolated rabbit (Christoph et al., 2017, 2018; Christoph & Luther, 2018) and pig (Christoph et al., 2018) hearts and a combination of marker-free motion tracking and ratiometry was applied to rabbit hearts (Kappadan et al., 2020). Recently, GPU-accelerated motion tracking has been successfully applied to isolated hearts of multiple species including mouse, rabbit and pig (Lebert et al., 2022).”

Referee #2:

The authors present a thorough review on optical mapping of contracting hearts. This review is well-structured and the necessary literature is well-cited. Please find my minor concerns below.

We thank the reviewer for their supportive comments and for their feedback.

(1) In the section on electromechanical uncoupler the authors write several times "BDM causes" or "Blebbistatin may affect". Is it certain that these agents are causal? The design of studies investigating this matter are often such that the presence of an uncoupler is compared with the absence of the uncoupler. However, the absence of the uncoupler results in contraction, which causes e.g. high energy demand, pressure changes and stretch. Could it be that many

of the differences observed between with and without uncoupler are actually differences caused by beating and non-beating?

Thank you for this important comment. We totally agree with the fact that observed effects might just be the difference between the beating and non-beating hearts as the metabolic demand and oxygen consumption are higher in beating hearts. To provide an unbiased overview, this issue with the metabolic challenges is described (**Page 7, lines 171-178** as follows:

*“However, we also observed morphology changes in OAPs indicating possible ischemia during high frequency electrical stimulations (**Figure 4D**, Kappadan et al., 2020). This is because metabolic demand and oxygen consumption of contracting hearts are higher as compared to hearts administered with blebbistatin and hence contracting hearts are more prone to ischemia especially while pacing at higher frequencies. Therefore, the observed differences may not be the direct effects of blebbistatin but may be an indirect consequence of the altered metabolic state of the heart when the contraction is inhibited using an excitation-contraction uncoupler.”*

(2) In lines 179-182 the authors write: "As the heart contracts, a camera pixel that was previously recording signals from one region of the heart will now be recording signals from another region, which can have very different fluorescence intensity. These variations in fluorescence "can be larger than the intensity change due to action potential, thus distorting the OAP signal while the heart moves." I am convinced that this is the main explanation for movement artifacts in optical action potentials. This because recordings of optical calcium transients show way less movement artifacts than optical action potentials from the same location. The main difference is the baseline fluorescence (during diastole) which is very high for voltage dyes but not for calcium sensors. The authors could considers discussing these observations.

Contraction of myocytes results in higher membrane density per pixel resulting in a higher signal amplitude and could contribute to movement artifact in the optical action potentials.

Thank you for your feedback. These points are incorporated (**Page 9, lines 222-226 and lines 230-232**) while discussing the factors contributing to motion artifacts as:

“For example, baseline fluorescence (during diastole) of calcium-sensitive dyes is less as compared to voltage-sensitive dyes results in lesser motion artifacts in optical calcium transients in comparison with optical action potential. Temporal variation of dye concentration due to photobleaching will also cause motion artifacts by affecting the baseline fluorescence.”

“Furthermore, contraction of cardiomyocytes results in higher membrane density per pixel resulting in a higher signal amplitude and could potentially contribute to motion artifacts in the OAPs.”

(2) Line 222. Consider writing "frames" instead of "video frames".

Thank you. "video frames" is replaced by "frames" (Page 10, line 275)

(3) Line 405 "lie" instead of "lies".

Thank you. We have corrected this (Page 16, line 483).

(4) In lines 407-409 the authors write "Hence, motion artifacts that originate from the inhomogeneous illumination of the tissue are not common in the recorded OAPs, which makes elimination of illumination artifacts difficult to perform." This sentence is not clear to me. Maybe the authors could adjust to sentence to make it clearer.

Thank you. The sentence has been modified to make it clearer (Page 16, lines 485-487):

"Hence, illumination-related motion artifacts are not common in the recorded OAPs and ratiometry will not remove illumination artifacts completely."

The authors describe several motion tracking techniques that can be used for correcting optical signals for movement artifacts. In order to check whether the correction has worked the corrected optical action potential should be compared with a microelectrode recording or, at least, the RT80 should coincide with the upslope of the T-wave of an unipolar electrogram. I encouraged the authors to discuss this matter and use a figure to illustrate this validation.

Thank you for the suggestion. A paragraph (Page 11, lines 310-317) is added to discuss the validation of motion tracking as follows:

*"Even though marker-free motion tracking enables to perform optical mapping experiments on isolated contracting hearts, it is important to verify the accuracy of the technique to avoid tracking error and data misinterpretation. The study by Christoph and Luther validated the robustness and accuracy of 2D motion tracking using experimental and synthetically generated optical mapping videos (Christoph & Luther, 2018). They achieved considerable reduction (75-80 %) in motion artifacts while comparing the tracked data with simulated ground-truth data. Furthermore, validation of the technique was performed using microelectrode recording as shown in **Figure 3D**."*

Additionally, a subfigure is added (**Figure 3D**) to show the comparison between a motion tracked optical action potential signal with a microelectrode recording.

The authors write almost 1,5 pages about the limitations of optical mapping of contracting hearts. Still in the summary section they advocate the use of this approach. I encourage the authors to be more precise in explaining in which situations optical mapping of contracting hearts is required. But also, to explain when the use of mechano-electrical uncouplers is sufficient or even

preferred.

Thank you for the feedback. A paragraph is added in the summary (**Pages 18-19, lines 552-561**) to explain the situations where optical mapping of contracting heart is required over electromechanically uncoupled conditions.

“However, it is important to note that isolated contracting heart experiments can also be compromised in certain situations owing to the higher oxygen and metabolic demands. In such condition, excitation-contraction uncoupling using blebbistatin may provide more accurate results and can be preferred. However, on the contrary, there could be effects such as ischemia underestimation while using blebbistatin due to lower metabolic and oxygen demands. Therefore, there is no single ideal experimental technique to use in all scenarios. Hence, the decision on whether to use contracting hearts or EC uncoupled heart depends on the specific research goals. For measurements involving metabolism and cardiac electromechanics including MEF, we recommend using contracting heart optical mapping.”

Dear Dr Ng,

Re: JP-TR-2022-283683R1 "Optical mapping of contracting hearts" by Vineesh Kappadan, Anies Sohi, Ulrich Parlitz, Stefan Luther, Ilija Uzelac, Flavio Fenton, Nicholas S Peters, Jan Christoph, and Fu Siong Ng

Thank you for submitting your manuscript to The Journal of Physiology. It has been assessed by a Reviewing Editor and by 1 expert referee and you are now invited to respond to the comments of the reviewers and submit a revised version for further consideration.

ABSTRACT FIGURES: Authors may use The Journal's premium BioRender account to create/redraw their Abstract Figures (and any other suitable schematic figure). Information on how to access this account is here: <https://physoc.onlinelibrary.wiley.com/journal/14697793/biorender-access>.

REVISION CHECKLIST:

We look forward to receiving your revised submission.

Yours sincerely,

Professor Laura Bennet
Senior Editor
The Journal of Physiology
<https://jp.msubmit.net>
<http://jp.physoc.org>
The Physiological Society
Hodgkin Huxley House
30 Farringdon Lane
London, EC1R 3AW
UK
<http://www.physoc.org>
<http://journals.physoc.org>

EDITOR COMMENTS

Reviewing Editor:

This revised manuscript by Kappadan et al. reviews recent advances and challenges of Optical Mapping (OM), with a focus on contracting hearts.

While the manuscript has been substantially revised, there are still some issues which would need to be addressed. The authors should be objective and be mindful of potential bias. Also, they should discuss the feasibility of OM beating hearts more than the effect of EC uncouplers, as the latter is not the focus of the review. This would greatly enhance the novelty and impact of this work.

Senior Editor:

Thank you for your revisions. The reviewer and editor have highlighted remaining significant concerns that must be addressed for the review to meet the threshold for further consideration. Greater rigour is required with referencing, and the review remains perceived as biased. The authors need to strongly reflect on their statements with an eye to bias, and remove such statements or tone them down. Attention is also required to providing greater clarity and to accuracy of referencing.

For your convenience the TR guidelines are : Topical Reviews, either as stand-alone reviews or as part of a special issue on a broad topic of current interest. Topical Reviews should provide a succinct and accessible synthesis of current information in rapidly-developing areas of physiology. Authors should be forward-looking and present new questions for future research/developments and are encouraged to express their own opinion on a subject area and may be controversial if they wish to be, as science often moves fastest when ideas are challenged. However, Topical Reviews should still present a balanced view of the topic.

REFEREE COMMENTS

Referee #1:

Even the authors substantially revised the main text, several significant concerns still exist.

The abstract still has very strong and bias statements, which must be removed: Lines 30-33 "However, the application of excitation-contraction uncouplers not only makes the experimental preparations less physiological, but these agents have also been shown to have direct effects on myocardial electrophysiology, which can alter the results and conclusions of the experiments."

As the authors already acknowledged that depends from the experimental protocols blebbistatin may not have no any effects on the cardiac physiology (Page 7, lines 171-183) Therefore, the observed differences may not be the direct effects of blebbistatin but may be an indirect consequence of the altered metabolic state of the heart when the contraction is inhibited using an excitation-contraction uncoupler. It is therefore unclear if blebbistatin significantly affects cardiac

electrophysiology. To date, no studies have shown clear effects of blebbistatin on ion channels responsible for cardiac repolarization. Furthermore, incorrect usage of blebbistatin may lead to blebbistatin precipitation and alterations of cardiac electrophysiology (Swift et al., 2012) and may account for some of the observed electrophysiological changes following blebbistatin administration (Swift et al., 2021).'

the term "less physiological" should be avoided as the specific criteria for "physiological" were not provided.

To increase the impact on the optical mapping research as well as the originality of the review, the authors should revise the abstract and manuscript with more focus on the feasibility of OM beating hearts (e.g. computer vision algorithms and ratiometric techniques) rather than effects of EC uncouplers, which were previously described in other reviews (e.g. "Getting It Right" perspective by Swift et al 2021)

In the response to previous question #10 the authors provide a very general response didn't clarify chamber- and surface specific applications. No proof was provided that the approaches can be successfully applied for the highly trabeculated endocardium or the complex atrial structure.

Please acknowledge and discuss the feasibility of the computer vision algorithms and ratiometric techniques to map different animal models/species and cardiac chambers (e.g. ventricle or atria), smooth epicardial surface or the highly trabeculated endocardium

The authors should acknowledge the in-vivo to ex-vivo canine optical mapping study with near infrared dye di-4-ANBDQBS (Hansen et al Circulation: Arrhythmia and Electrophysiology. 2018;11:e006870) showed that sinus nodal activation, atrial conduction and AF activation pattern are very similar between in-vivo atrial mapping and ex-vivo atrial mapping with blebbistatin.

Table 1 "Effects of excitation-contraction uncouplers on cardiac electrophysiology" is still confusing and it is not directly relevant to the goal of the review (OM of beating hearts). It should be replaced on the Table, which would provide the feasibility and pros/cons of "Techniques to reduce motion artifacts" to map different animal models/species and cardiac chambers (e.g. ventricle or atria), smooth epicardial surface or trabeculated endocardium.

The current Table is still missing species (e.g. rabbit, canine, human...) as well as chambers (e.g. atria or ventricle)

In the response to my previous question #9, the authors provided Line 85-88 "However, the effects of near infrared dyes such as di-4-ANBDQBPQ (JPW-6003) and di-4-ANBDQBS (JPW-6033), designed for blood perfused optical mapping (Matiukas et al., 2007), have not been reported or studied in detail.'

In fact, the in-vivo and ex-vivo effects of di-4-ANBDQBS were studied reported in Lee et al 2019 Cardiovascular Research (Figure 7 shows that the dye doesn't have any significant cardiac acute and 10 days toxicity in the pig mode)

REQUIRED ITEMS:

Please upload your author profile(s) as a Word file.

END OF COMMENTS

Response to reviewers: Optical mapping of contracting hearts (JP-TR-2022-283683R1)

Thank you for the positive outlook on our manuscript entitled “Optical mapping of contracting hearts”, and for providing thoughtful comments and suggestions for improving our manuscript.

We appreciate the opportunity to revise our work, and we are pleased to hereby submit a revised version of our manuscript for possible publication in *The Journal of Physiology*. We have comprehensively addressed all points raised by the reviewers and provide a detailed point-by-point response below. The reviewers' comments are in **bold italics** and are followed by the response to the comments and the revisions undertaken. We have highlighted the changes made to the manuscript in response to these comments (**Page and Line numbers** below refer to the Redlined document). As a result of the incisive comments, we believe the manuscript has significantly improved. We hope the revised manuscript is in line with the editor's expectation as well as those of the reviewers.

EDITOR COMMENTS

Reviewing Editor:

This revised manuscript by Kappadan et al. reviews recent advances and challenges of Optical Mapping (OM), with a focus on contracting hearts.

While the manuscript has been substantially revised, there are still some issues which would need to be addressed. The authors should be objective and be mindful of potential bias. Also, they should discuss the feasibility of OM beating hearts more than the effect of EC uncouplers, as the latter is not the focus of the review. This would greatly enhance the novelty and impact of this work.

We thank the Reviewing Editor for the feedback. The manuscript has been carefully revised to avoid any potential bias. The feasibility of contracting heart optical mapping and motion correction techniques for multiple animal species and chambers are described in detail and the section on the effect of EC uncouplers is substantially modified to provide an unbiased overview.

Senior Editor:

Thank you for your revisions. The reviewer and editor have highlighted remaining significant concerns that must be addressed for the review to meet the threshold for further consideration. Greater rigour is required with referencing, and the review remains perceived as biased. The authors need to strongly reflect on their statements with an eye to bias, and remove such statements or tone them down. Attention is also required to providing greater

clarity and to accuracy of referencing.

We thank the Senior Editor for the feedback. The concerns highlighted by the reviewer and editor have been carefully addressed and the manuscript is revised to remove any potential bias. For example, the **Table** has been completely removed to avoid further confusion or bias.

Proper attention has also been given to the accuracy of referencing.

REFEREE COMMENTS

Referee #1:

Even the authors substantially revised the main text, several significant concerns still exist.

The abstract still has very strong and bias statements, which must be removed: Lines 30-33 "However, the application of excitation-contraction uncouplers not only makes the experimental preparations less physiological, but these agents have also been shown to have direct effects on myocardial electrophysiology, which can alter the results and conclusions of the experiments."

As the authors already acknowledged that depends from the experimental protocols blebbistatin may not have no any effects on the cardiac physiology (Page 7, lines 171-183) Therefore, the observed differences may not be the direct effects of blebbistatin but may be an indirect consequence of the altered metabolic state of the heart when the contraction is inhibited using an excitation-contraction uncoupler. It is therefore unclear if blebbistatin significantly affects cardiac electrophysiology. To date, no studies have shown clear effects of blebbistatin on ion channels responsible for cardiac repolarization. Furthermore, incorrect usage of blebbistatin may leads to blebbistatin precipitation and alterations of cardiac electrophysiology (Swift et al., 2012) and may account for some of the observed electrophysiological changes following blebbistatin administration (Swift et al., 2021).'

the term "less physiological" should be avoided as the specific criteria for "physiological" were not provided.

We thank the Referee for the feedback. The term "less physiological" is removed from the abstract. The abstract is revised to provide an unbiased overview of the topic (**Page 2, lines 30-31**) as follows:

“However, such experimental preparations eliminate the possibility of electromechanical interaction and effects such as mechano-electric feedback (MEF) cannot be studied.”

To increase the impact on the optical mapping research as well as the originality of the review, the authors should revise the abstract and manuscript with more focus on the feasibility of OM beating hearts (e.g. computer vision algorithms and ratiometric techniques) rather than effects of EC uncouplers, which were previously described in other reviews (e.g. "Getting It Right" perspective by Swift et al 2021)

Thank you for the feedback. We have revised the manuscript to avoid any possible bias. The section *“Disadvantages of optical mapping of non-contracting hearts”* is completely removed and two paragraphs are added instead into the section *“Optical mapping of non-contracting hearts”*. The first paragraph provides an unbiased overview of EC uncouplers (Page 6, lines 133-151) and the second paragraph talks about the importance of performing contracting heart optical mapping (Pages 6-7, lines 152-159) as follows:

“Although the excitation-contraction uncouplers are designed to suppress contraction while preserving the electrical activity, BDM and Cyto-D were associated with electrophysiology changes in multiple studies (Riccio et al., 1999; Lee et al., 2001; Banville & Gray, 2002; Kettlewell et al., 2004; Pitruzzello et al., 2007). Blebbistatin, instead, has been considered the electromechanical uncoupler of choice since it was introduced in 2003 (Straight et al., 2003) and is considered to have minimal effects on cardiac electrophysiology (Fedorov et al., 2007, 2009; Dou et al., 2007; Fenton et al., 2008; Jou et al., 2010; Lou et al., 2012; Hansen et al., 2018). Contrastingly, a few studies have recently reported effects of blebbistatin on cardiac electrophysiology (Brack et al., 2013; Lee et al., 2019; Kappadan et al., 2020). The reason behind this disparity could be differences in the metabolic demand and oxygen consumption of contracting hearts as compared to hearts administered with blebbistatin (S et al., 2015; Swift et al., 2021). Therefore, the observed differences may not be the direct effects of blebbistatin but may be an indirect consequence of the altered metabolic state of the heart when the contraction is inhibited using an excitation-contraction uncoupler. To date, no studies have shown clear effects of blebbistatin on ion channels responsible for cardiac repolarization. Furthermore, incorrect usage of blebbistatin may lead to blebbistatin precipitation and alterations of cardiac electrophysiology (Swift et al., 2012) and may account for some of the observed electrophysiological changes following blebbistatin administration (Swift et al., 2021).

Even though the excitation-contraction uncouplers allow recording of optical action potentials without motion artifacts, crucial information contained in cardiac

contractility and coupled cardiac electromechanics are lost. Moreover, electromechanical coupling is bidirectional such that electrical excitation of the cardiac cell membrane causes mechanical contraction via excitation-contraction coupling (ECC) and mechanical stretch on the myocardium can cause electrical excitation through stretch activated channels via a process known as mechano-electric feedback (MEF) (Franz, 1996, 2000; Taggart, 1996; Zabel et al., 1996), and such effects cannot be studied in a non-contracting heart.”

Additionally, feasibility of contracting heart optical mapping techniques to map different animal species and cardiac chambers are also described (Pages 17, lines 493-511) as an answer to question #10 (see below).

In the response to previous question #10 the authors provide a very general response didn't clarify chamber- and surface specific applications. No proves were provided that the approaches can be successfully applied for the highly trabeculated endocardium or the complex atrial structure.

Please acknowledge and discuss the feasibility of the computer vision algorithms and ratiometric techniques to map different animal models/species and cardiac chambers (e.g. ventricle or atria), smooth epicardial surface or the highly trabeculated endocardium

Thank you for your feedback. The paragraph is now modified to discuss the feasibility of the motion tracking and ratiometric techniques to map different animal species and cardiac chambers as (Page 17, lines 493-511) follows:

“It is also important to discuss the feasibility of ratiometric and marker-free motion tracking techniques to map different animal species and cardiac chambers. The ratiometric optical mapping approach has been successfully used to minimize motion artifacts from OAPs of isolated rabbit (Knisley et al., 2000; Dumas & Kinisley, 2005; Kappadan et al., 2020) and pig (Bachtel et al., 2011; Bourgeois et al., 2011; Lee et al., 2019) whole hearts. Ratiometry was also employed to record action potentials from neonatal rat ventricular myocytes (NRVMs) (Chowdhury et al., 2018) and human induced pluripotent stem cell-derived cardiomyocytes (hiPSC-CM) (Hortigon-Vinagre et al., 2016). Moreover, ratiometry utilizing calcium sensitive dye, Indo-1 was used to suppress motion artifacts from intracellular Ca^{2+} concentrations in isolated rat hearts. The marker-free motion tracking has been already used in isolated rabbit (Christoph et al., 2017, 2018; Christoph & Luther, 2018) and pig (Christoph et al., 2018) hearts and a combination of marker-free motion tracking and ratiometry was applied to isolated rabbit hearts (Kappadan et al., 2020). The motion tracking in these studies were performed on the optical mapping videos of whole heart (atria and ventricles) with a focus on ventricles during pacing and fibrillation. Recently, GPU-accelerated motion tracking has been successfully applied to hiPSC-CM and isolated hearts of mouse, rabbit and pig (Lebert et al., 2022). However, to our knowledge, no

studies have performed motion correction specific to highly trabeculated endocardium or complex atrial structure.”

The authors should acknowledge the in-vivo to ex-vivo canine optical mapping study with near infrared dye di-4-ANBDQBS (Hansen et al Circulation: Arrhythmia and Electrophysiology. 2018;11:e006870) showed that sinus nodal activation, atrial conduction and AF activation pattern are very similar between in-vivo atrial mapping and ex-vivo atrial mapping with blebbistatin.

Thank you for your feedback. We have acknowledged the study by Hansen et al. (Page 6, lines 136-140) as follows:

“Blebbistatin, instead, has been considered the electromechanical uncoupler of choice since it was introduced in 2003 (Straight et al., 2003) and is considered to have minimal effects on cardiac electrophysiology (Fedorov et al., 2007, 2009; Dou et al., 2007; Fenton et al., 2008; Jou et al., 2010; Lou et al., 2012; Hansen et al., 2018).”

Table 1 "Effects of excitation-contraction uncouplers on cardiac electrophysiology" is still confusing and it is not directly relevant to the goal of the review (OM of beating hearts). It should be replaced on the Table, which would provide the feasibility and prone/cons of "Techniques to reduce motion artifacts" to map different animal models/species and cardiac chambers (e.g. ventricle or atria), smooth epicardial surface or trabeculated endocardium.

The current Table is still missing species (e.g. rabbit, canine, human...) as well as chambers (e.g. atria or ventricle)

Thank you for your feedback. We have removed the Table completely to avoid any confusion or bias.

In the response to my previous question #9, the authors provided Line 85-88 "However, the effects of near infrared dyes such as di-4-ANBDQPQ (JPW-6003) and di-4-ANBDQBS (JPW-6033), designed for blood perfused optical mapping (Matiukas et al., 2007), have not been reported or studied in detail.'

In fact, the in-vivo and ex-vivo effects of di-4-ANBDQBS were studied reported in Lee et al 2019 Cardiovascular Research (Figure 7 shows that the dye doesn't have any significant cardiac acute and 10 days toxicity in the pig mode)

Thank you for your feedback. We have added the information regarding the effect of di-4-ANBDQBS (Page 5, lines 90-93) as:

“However, intracoronary injection of the dye di-4-ANBDQBS (JPW-6033), designed for blood perfused optical mapping (Matiukas et al., 2007) showed no signs of cardiac toxicity during continuous monitoring of ECG from baseline to 30 minutes after dye/solvent injection in pig models. Moreover, heart rate was stable throughout the recording period (Lee et al., 2019).”

Dear Dr Ng,

Re: JP-TR-2023-283683R2 "Optical mapping of contracting hearts" by Vineesh Kappadan, Anies Sohi, Ulrich Parlitz, Stefan Luther, Ilija Uzelac, Flavio Fenton, Nicholas S Peters, Jan Christoph, and Fu Siong Ng

We are pleased to tell you that your paper has been accepted for publication in The Journal of Physiology.

Authors should note that it is too late at this point to offer corrections prior to proofing. The accepted version will be published online, ahead of the copy edited and typeset version being made available. Major corrections at proof stage, such as changes to figures, will be referred to the Editors for approval before they can be incorporated. Only minor changes, such as to style and consistency, should be made at proof stage. Changes that need to be made after proof stage will usually require a formal correction notice.

All queries at proof stage should be sent to: TJP@wiley.com

Yours sincerely,

Professor Laura Bennet
Senior Editor
The Journal of Physiology
<https://jp.msubmit.net>
<http://jp.physoc.org>
The Physiological Society
Hodgkin Huxley House
30 Farringdon Lane
London, EC1R 3AW
UK
<http://www.physoc.org>
<http://journals.physoc.org>

P.S. - You can help your research get the attention it deserves! Check out Wiley's free Promotion Guide for best-practice recommendations for promoting your work at www.wileyauthors.com/eoo/guide. You can learn more about Wiley Editing Services which offers professional video, design, and writing services to create shareable video abstracts, infographics, conference posters, lay summaries, and research news stories for your research at www.wileyauthors.com/eoo/promotion.

IMPORTANT NOTICE ABOUT OPEN ACCESS: To assist authors whose funding agencies mandate public access to published research findings sooner than 12 months after publication The Journal of Physiology allows authors to pay an Open Access (OA) fee to have their papers made freely available immediately on publication.

You can check if your funder or institution has a Wiley Open Access Account here: <https://authorservices.wiley.com/author-resources/Journal-Authors/licensing-and-open-access/open-access/author-compliance-tool.html>

EDITOR COMMENTS

Reviewing Editor:

Thank you for your revision of "Optical mapping of contracting hearts" by Kappadan et al. All issues have now been addressed including the potential bias of the authors. The quality of the manuscript has significantly improved and therefore

is likely to be a valuable resource for researchers working in the field of cardiovascular science and optical mapping.

REFeree COMMENTS

Referee #1:

The authors have addressed all my previous questions. The review has significantly improved as a result of the revisions. Congratulations!

2nd Confidential Review

09-Feb-2023